

# Evolution of squall line variability and error growth in an ensemble of LES

Edward Groot[1] and Holger Tost[1]

[1]Institut für Physik der Atmosphäre, Johannes Gutenberg Universität, Johannes-Joachim-Becher-Weg 21, Mainz, Germany

**Correspondence:** Edward Groot (egroot@uni-mainz.de)

**Abstract.**

Squall lines represent an organized form of atmospheric convection that link processes occurring at the small end of the mesoscale and processes ocurring at the large end of the mesoscale. This study analyses the initial condition sensitivity of idealized squall lines in an LES ensemble. The ensemble spread of the squall lines is evaluated using passive tracers, an ensemble sensitivity analysis, other statistical tools and an error growth metric. Analysing gravity wave dynamics, convective initiation, squall line relative motion and updraft/downdraft characteristics and transport, a chain of interacting processes is identified.

From the convective point of view ensemble spread is rooted in a secondary phase of convective initiation (30-35 min) a few km ahead of the squall line. Contrasts in the amount secondary initiation arise within the ensemble, as vertical velocity varies at the location of convective initiation within the ensemble due to differences in gravity wave amplitude and phase. Immediately after the secondary phase of initiation (30-45 min), the cold pool accelerates to velocities of 2-4 m/s (ensemble envelope).

With the spread in secondary convective initiation, upward mass transport is disturbed, which also affects downdraft mass fluxes. Furthermore, once accelerated (30-40 minutes), the cold pool nearly maintains its propagation speed in each ensemble member. It is shown that part of the errors occurring after 45-85 minutes are explained by the cold pool velocity and a correction for cold pool velocity removes a substantial fraction of the spread. A coherent anomaly of the circulation within the squall line, which is consistent with extra upward mass transport, exists during this phase of the evolution. It is proposed that the identified chain of interactions may be explained by a common mode of variability, which determines a substantial portion of the ensemble spread in the stage after 30-85 minutes in many diagnostics.

Based on a non-monotonic relation between initial conditions and local vertical velocities that cause secondary initiation, one can argue that an intrinsic limit of predictability exists, as Melhauser and Zhang (2012) do.

## 1 Introduction



A squall line is a complex meteorological phenomenon consisting of an elongated linear area with convective cells that usually induce a coherent mesoscale circulation. It often develops into a mixed stratiform-convective precipitation system, growing to about a hundred or several hundred kilometers in the horizontal and living for several hours or longer. They have been an active research area over many decades (Houze, 2004, 2018) .

Increasing computational power has lead to increasingly high resolution numerical simulations in convective studies. Through the last 25 years, it has been possible to go from highly simplified cloud resolving km-scale (Weisman et al., 1997) to systematic sets of fully three dimensional large eddy simulations (LES) at high resolutions (Adams-Selin, 2020a, b). Much work has been done on understanding the impact of differences in shear profiles and its consequence for convective organisation (e.g. Weisman and Rotunno, 2004; Coniglio et al., 2006; Adams-Selin, 2020a), including documentation of squall line organisation. Moreover, the effect of resolution on squall line evolution has been an active topic (Weisman et al., 1997; Bryan et al., 2003; Lebo and Morrison, 2015), where the question whether squall line simulations converge at high resolution has been investigated. Bryan et al. (2003) found that LES simulations do not result in true convergence of solutions, but still resolving at the hundred-meter scales strongly increases the confidence in squall line simulations compared to 1-4 km simulations, as the needs for parameterising processes shifts to subgrid turbulence in a partly represented inertial subrange and microphysics. How squall lines depend on microphysics, shear and instability can be investigated rigorously by now, (e.g. Morrison et al., 2009; Grant et al., 2018; Adams-Selin, 2020a, b).

Research focused on squall line ensembles and their error growth has been carried out by Melhauser and Zhang (2012); Hanley et al. (2013); Weyn and Durran (2017). Ensemble members were compared to each other with a focus on the differences in their evolution.

Melhauser and Zhang (2012) and Hanley et al. (2013) looked at a set of simulations featuring a real squall line case in a local area model from the larger mesoscale and synoptic point of view. This was the core feature of both sensitivity studies, even though downscale "contamination" from the synoptic scale was present and considered. Melhauser and Zhang (2012) found that an intrinsic limit of predictability can affect squall line ensemble forecasts: by reducing initial condition spread by a factor of 8, their simulations could still diverge about as much as with unreduced initial condition spread. Furthermore, the contingency of convective initiation was found to be a key for the ensemble spread. Some very similar results were found by Hanley et al. (2013).

Weyn and Durran (2017) have looked at error growth in mesoscale convective systems with squall line like features in isolation. Their simulations had a resolution of 1 km on a domain with horizontal scales of about 500 km. Most of their error growth analysis was done from the spectral point of view, which implies that there was much less attention on convective and accompanying processes in physical space. Despite this focus they have compared the error growth of divergent and rotational wind components and found that divergent winds are mostly affecting larger scales errors. Furthermore, an important finding was that by reducing initial condition spread by factors of 5 and 25, only about 1 hour of predictability was won. Compared to their saturation time scale of about 5 hours this is not much.

In this study, sensitivity analyses will be carried out from a dynamical point of view with squall line simulations in isolation





from a synoptic and larger mesoscale environment. Error growth will be analysed in high resolution simulations (200 m) with 10 ensemble members. The dynamical aspects that will be analysed include the following:

– **Gravity wave dynamics**

Interaction between gravity wave dynamics and a convective environment has been investigated using a linear gravity wave model (Bretherton and Smolarkiewicz, 1989; Nicholls et al., 1991; Mapes, 1993). The latter argued that propagation of the gravity waves is important for upscale organisation of convective systems as it happens in the tropics. In these studies a heating signal was used as a proxy for a convective system, mimicking the latent heating source. Subsequently, cloud models have been used to investigate how gravity wave dynamics can concentrate regions of upward motion, with several vertical gravity wave modes favoring certain regions of convective initiation that can subsequently assist to form coherent patterns and upscale organisation (e.g. Lane and Reeder, 2001; Stechmann and Majda, 2009; Adams-Selin and Johnson, 2013; Lane and Zhang, 2011; Grant et al., 2018, and references in the latter two). In addition, Bierdel et al. (2017) have created a model for better understanding of error propagation from convective scales to large mesoscales and (sub-)synoptic scales caused by differences in convective heating based on the linear gravity wave model (Bretherton and Smolarkiewicz, 1989; Nicholls et al., 1991), but focusing on the role of rotation on these scales and geostrophic adjustment of difference flow. That scale is substantially larger than (sub-)squall line scales investigated here.

– **Convective initiation**

Looking from the convective initiation point of view, Fovell et al. (2006) showed how gravity waves affect squall line regions on smaller scales. In their simulations with explicitly resolved deep convection it was shown how gravity waves can trigger initiation of discrete convective cells ahead of a squall line, that can subsequently merge with the main system. A sensitivity of these discrete convective cell was identified, which lead to a dependence of initiation on the active treatment of radiation. The sensitivity of convective initiation has played an important role in Melhauser and Zhang (2012); Adams-Selin (2020a, b) and can nowadays be resolved more accurately due to finer grid spacing.

– **Cold pool relative motion, or more specifically $u$ in x-z plane averaged along squall line**

Pandya and Durran (1996) have looked at the (sub)system scale variability explained by thermally forced gravity waves and found that the mesoscale squall line circulation in their system depends to a large extent on the magnitude and shape of thermal forcing. Low level features, notably the rear inflow jet, depend on the low level conditions, notably $N^2$. Similarly to their study, this study has a strong focus on the squall line system circulation ($u$) in the plane perpendicular to the squall line orientation. Moreover, squall line sensitivity is explored with ensemble perturbations, but the physical processes are not modified here. Whereas the Pandya and Durran (1996) study has a focus on the major features of the convective system, our focus is also on sensitive dependence on small scale dynamics as affected by initial conditions.

– **Updraft and downdraft motion**

Updraft and downdraft characteristics can be sensitive to resolution and that may lead to biases in updrafts (or downdrafts) when a comparison to radar images is made (Varble et al., 2020; Lebo and Morrison, 2015; Varble et al., 2014,



e.g.), which could propagate to other aspects of squall line simulations (Varble et al., 2020, 2014). However, since no
comparison between simulations at different resolutions or observations and simulations is made here, these are not
applicable. Furthermore, the grid spacing is believed to be fine enough for accurately resolving updraft and downdraft
characteristics (Bryan et al., 2003; Lebo and Morrison, 2015; Varble et al., 2020). Nevertheless, updrafts and downdrafts
will be detected and their role in squall line dynamics is investigated.

The main limitations of this study are that just one idealized squall line case with weak ensemble perturbations is considered and
that the squall line evolution is initially nearly 2D due to nearly 2D flow and initial conditions. However, this nearly 2D structure
is of benefit for the analysis of squall line relative flow. Considering just one case also allows for detailed consideration of the
dynamical aspects listed. The magnitude of ensemble perturbations that are applied are equivalent to a vertical wind profile
uncertainty of about or less than one model layer in the vertical in typical convection permitting models. Altogether the aim is
to get an comprehensive overview of the processes in which errors are showing up in a high resolution squall line ensemble,
following such a small initial condition perturbation. We also describe how these errors may be transferred from one process
to another. Furthermore, intrinsic predictability can be addressed, given the small magnitude of initial condition perturbations.
Zonal shear in a shallow layer is perturbed with magnitudes of 0-5% and randomly, as opposed to the often used systematic
iterative approaches (e.g. Selz, 2019; Zhang et al., 2019; Selz et al., 2022; Melhauser and Zhang, 2012).

Techniques are applied of which some are not so commonly used in studies of mesoscale convective systems. Next to more
widely used passive tracers (e.g. Grant et al., 2018), an ensemble sensitivity analysis (Hanley et al., 2013; Torn and Romine,
2015; Bednarczyk and Ancell, 2015, but the latter with parameterized convection) and a difference kinetic energy metric are
used. The difference kinetic energy metric is of particular interest for error growth analysis (Zhang, 2005; Zhang et al., 2007).
Passive tracers are an important tool to identify variability in convective transport, but by targeting at an inflow layer, contrasts
can be identified early on. Variability in a secondary phase of convective initiation is revealed this way. In combination with
tailored ensemble sensitivity analysis which allows one to connect the strength of the convective initiation with subsequent
evolution of the squall line circulation ($u$), the effects of this phase of initiation can be followed in time. System relative
flow will be evaluated and its relation with secondary convective initiation is investigated. The difference kinetic energy error
growth metric and specific diagnostics can provide further insights in the evolution of system relative motion and its role in
error growth.

In Section 2 the characteristics of the simulations are documented, as well as the initial conditions and ensemble design.
Furthermore, the general region and time windows of focus are pointed out and the statistical verification techniques are
described. In Section 3 the main analysis is carried out, preceded by a description of diagnostics used in a subsection. The
secion starts with a general evolution of the simulated squall line echoes. After looking at the simulated radar reflectivities, the
comparison section (Section 3.2) describes secondary convective initiation and identifies a relation with gravity wave signals.
This is followed by an investigation of the cold pool propagation (Section 3.3.1). Then ensemble sensitivity analysis (Section
3.3.2) assesses the connection with the squall line relative circulation, followed by an investigation of downdrafts and additional
statistical considerations (Section 3.3.3). Section 3 ends with an error growth (Section 3.4) analysis in grid point space and a



system relative flow feedback to highlight associated contrasts in error growth. The set of analyses is synthesized in Section 4,
followed by a discussion (also in that section). This discussion leads to the conclusions, as given in Section 5.

## 2 Methods

### 2.1 Model configuration

Experiments are conducted with cloud resolving model CM1 on a 120 by 120 km grid with a depth of 20 km and duration
of 2 hours simulation time, using the model version of Bryan (2019). The time steps are 0.75 s, with output stored every 5
minutes. All grid cells have a 200 m horizontal grid spacing and 100 m in the vertical. The upper 5 kilometers of the domain
consists of a sponge layer with damping that absorbs upward propagating gravity wave signals. CM1 is running in the large
eddy simulation mode, where the subgrid turbulence model is handled with a TKE-scheme after Deardorff (1980). The default
CM1 microphysics scheme is used, which is the 2 moment Morrison scheme with hail (Morrison et al., 2009). The advection
scheme applies 5th order advection and Coriolis acceleration is ignored. Radiation is not actively resolved. All simulations
have used the aforementioned settings.

The boundary conditions are non-periodic in all directions, which leads to the theoretical availability of an infinite reservoir of
inflow air. Derivatives of all quantities are set to 0 at the boundaries. Wave signals can therefore partially reflect at boundaries
and to reduce effects of boundary reflection, the outer regions on the north and south ends of the squall line are excluded from
the main analysis framework. More details and download instructions for the namelist file and output data of all simulations
can be found in the setup and namelist material at https://tinyurl.com/groot-tost-22 (Groot, 2022).

### 2.2 Environmental conditions and ensemble perturbations

The thermodynamic profile in each of the simulations is based on Weisman and Klemp (1982) (Figure 1), which represents
the conditions in the eastern half of the domain at $t = 0$ minutes. Below $z = 2500$ m, a potential temperature perturbation that
decreases linearly in magnitude with $z$ from -6 K at the surface to 0 at 2500 m is set at initial time for the western half of the
domain, $x < 0$ km. The contrast between these two air masses triggers upward motion at their interface ($x = 0$ km) and the
given environment leads to a convectively very unstable environment with a CAPE of about 2000 J/kg for rising warm air mass
in the east. The basic character of the wind profile is sheared and adjusted from Rotunno et al. (1988), with the u-component
linearly varying from 12.5 m/s easterly inflow at the surface to weak westerly flow of 1.5 m/s above the top of the shear layer
(Figure 1). That top of the shear layer is set at $z_{i,ref} = 2500$ m for the reference simulation. The v-component of the wind varies
linearly from -2 (surface) to +2 m/s ($z \geq z_i$) over the same layer. The latter is needed to develop some three dimensionality
in the simulation. Given fully 2D initial conditions, the three dimensionality could have been absent without any differential
meridional advection in combination with the open boundary conditions. Given that wind profile, the shear vectors are nearly
perpendicular to the initial cold pool boundary. In combination with the strong convective instability near the interface, it leads
to the formation of a squall line.



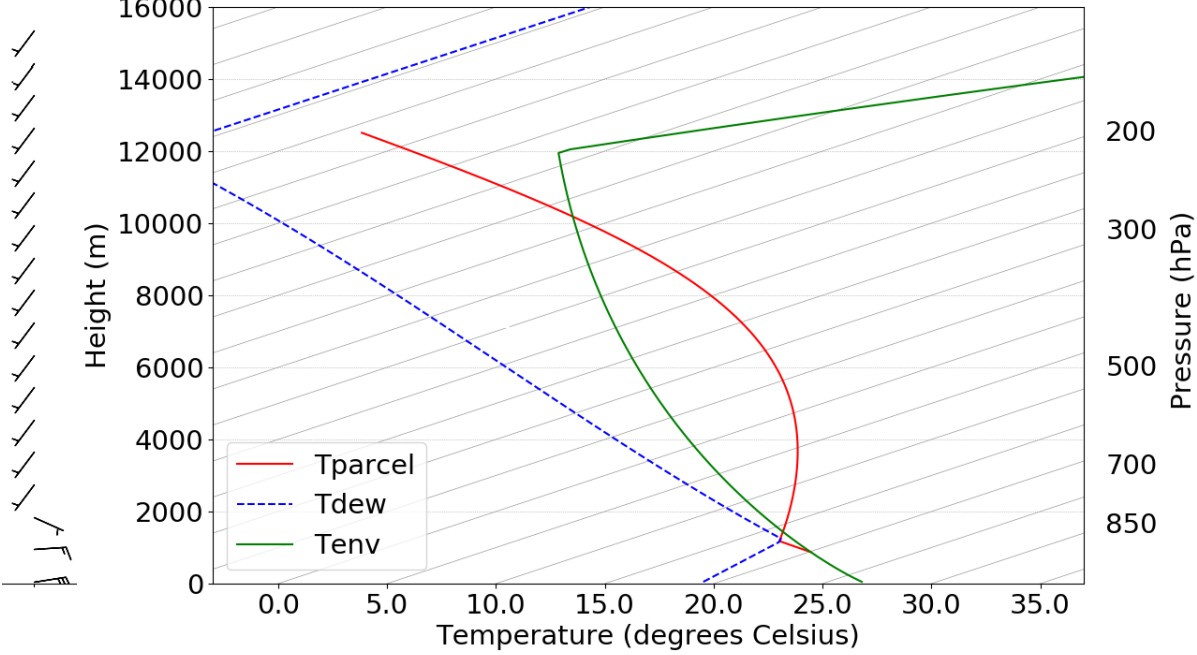

**Figure 1.** Thermodynamical profile based on Weisman and Klemp (1982) and wind profile after Rotunno et al. (1988), with adjustments (left). Temperature: green solid line; dew point: blue dashed line; convectively lifted undiluted parcel's temperature: red line.

Note that the wind profile implies the omission of deeper shear, which on average reduces the depth of vertical displacements and the size of convective systems (Coniglio et al., 2006). This is beneficial for a reduction of the storm relative flow and hence to reduce the across line growth of the squall lines: their residence time within the domain is increased.

An ensemble is generated by randomly perturbing the interface height ($z_i$), with a maximum absolute deviation from the reference height $z_{i,ref}$ of -127 m (5%) for ENS-05 $z_{i,ENS-05}$ among the 9 randomly drawn perturbed interface heights

$z_{i,ENS-X}$ and a mean absolute deviation of 67 m (2.7%) from $z_{i,ref}$ (see also Groot, 2022). That is equivalent to about one model level in high resolution models. Note that the cold pool depth and hence the potential energy distribution is not affected by the ensemble perturbations; the kinetic energy distribution of the intial conditions shows only weak deviations at low levels within the ensemble.

## 2.3    Spatial and temporal windows for diagnostics

Presented diagnostics are applied to a central portion of the squall line region, where $40 > y > -20$ km. This is to reduce the effects of the boundaries and their wave reflections on the analysis of the squall line evolution. Generally, boundary effects are only (clearly) noticeable in the central portion of the domain by the last 30-40 minutes of the simulation. Furthermore, the ensemble members all have slightly different boundary conditions, as controlled by their own evolution nearby/at the



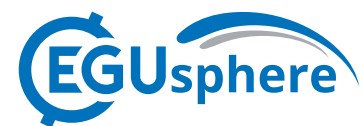

boundaries. The boundary conditions are solely based on their conditions, with the first derivatives set to zero right at the boundary. Therefore these are initially inherited by $z_i$ as much as their whole difference evolution to the control or ensemble mean is essentially controlled by the initial value of $z_i$.

In the x-direction a region within 1 km of the boundaries is excluded from the analysis.

Some analyses (ensemble sensitivity analysis; Section 3.3.2 and the error growth analysis of Section 3.4) is carried out relative to the *cold pool edge*, of which the detection is described in Section 3.3.1.

We emphasize the evolution up to 75-80 minutes of the simulations, as this part draws most of our attention. Nonetheless, the last 40 minutes of simulation time are not completely ignored.

### 2.4 Statistical assessment of the robustness of signals

A statistical test has to be defined to assess whether statistical analysis actually leads to a confident signal, especially given the small ensemble size: $n = 10$. With that ensemble size, correlation coefficients exceeding $|r| > 0.631$ are significant at $\alpha$ of 0.05. That means their frequency of occurrence in any large random sample is expected to be about 5% at $p = 0.05$ and 1.25% for random occurrence of $|r| > 0.75$ ($p = 0.0125$).

In this study, statistical grid point testing will be applied to spatial patterns within the squall line that contain or do not contain a statistically significant signal. Therefore it is expected that under the condition of a robust statistical effect within a certain feature of interest, this feature shows significant correlations over a large fraction of the grid points located within its area. In the limit case this fraction $f$ needs to theoretically fulfill a criterion close to $f > 2p$ (Wilks, 2016). On the contrary, other features without signals associated with it would simultaneously reveal small fractions of grid points with statistical significance. This could for instance be expected for most of the signals in the stratosphere, $z$ of $15 - 20$ km. If fully random, a significant area fraction $f$ of about 0.05 is expected, with $f \leq 2p = 0.1$ for $p = 0.05$ and 0.025 for $p = 0.0125$.

## 3 Results

### 3.1 Evolution of squall line radar reflectivity

As an exemplary overview of the general properties and behavior of squall line that develops in the ensemble of simulations, the evolution of the simulated radar reflectivity is discussed in the next section. It starts with the evolution of one member, followed by a comparison of two members at the outer ends of the ensemble envelope.

### 3.1.1 Evolution of a squall line, example simulation

As a response to the (1) large latent instability, (2) relatively strong low level wind shear perpendicular to the cold pool and the (3) strong forcing of upward motion by the cold pool, deep convection soon develops along the full length of the y-axis. Narrow echoes of about 40 dBz right above the dam breaking line at $x = 0$ km appear after 15 minutes into the simulation (Figure 2a). Five minutes later echoes exceeding 20 dBz reach above $z = 10$ km, as can be deduced from Figure 2c and 2d.





Echoes also widen and exceed 60 dBz in the core region of the line. For the following 30 minutes, the squall line echoes grow

200 monotonically at both anvil level and low/mid level. The increase in areas with reflectivity $> 20dBz$ occurs both upshear and

downshear of the convective core (as could be deduced from Figures 2g, h), which itself is more or less stationary above $x = 0$

km. However, then the growth halts for a while on the downshear side after about one hour and on the upshear side after about

75 minutes (Figure 2k). Some shrinkage of the areas with substantial cloud reflectivity occurs. After 80-90 minutes a stratiform

region of precipitation starts to develop on the upshear flank of the system (Figure 2j) and the anvil expands in both directions

toward the end of the simulation, growing to about 100 km length, which is nearly the full domain. However, the stratiform

precipitation area remains rather restricted to a very limited region at the rear flank (Figure 2j, at 3 km) and precipitation

intensities remain rather low.

Starting from initial conditions that depend on $x$ and $z$ but not on y, initially the system is obviously nearly homogeneous in $y$

(see Figure 2). The system gradually develops into a 3D squall line. Gradients develop along the y-axis with convective cells of

higher reflectivity embedded within the line after 50 minutes and beyond this time. However, we will mostly focus on results

averaged in the $y$-direction, as contrasts are largest in $x$- and $z$-directions and tend to be smoother along the y-axis.

### 3.1.2 Comparison of simulations: secondary phase of initiation

In Figure 2 the evolution of the squall line in ensemble member 3 is shown. From the convective point of view, this member is

on the very active end of the ensemble distribution. After 15 minutes (Figure 2a) the first convective cells are forming; after 25

minutes this initial line of cells, which is very homogeneous in the y-direction, is getting mature. So far, ensemble variation is

not yet noticeable.

Upon this mature stage of the first line of cells follows a phase were secondary initiation happens at $t = 30$ to 40 minutes.

This secondary phase of convective initiation is most distinctive in ensemble member 3, where a secondary line of cells is

triggered just ahead of the former line of cells and leads to forward displacements of the squall line core. The reflectivity at

$t = 50$ minutes clearly illustrates the forward displacement of convective cells and an increased area of reflectivity $> 55$ dBz on

the forward flank when the evolution of ensemble member 3 (ENS-03; Figure 2e) is compared with the reference run (Figure

2f). The reference run is basically not supporting the development of the secondary phase of initiation and hence the squall line

appears to be practically stagnant until $t = 50$ minutes (Figure 2f). If any of the echoes in the reference simulation reach $x > 5$

km, it is anvil precipitation with reflectivity that locally reaches about 40 dBz just below melting level, as opposed to newly

initiated updrafts with reflectivities reaching 55-65 dBz for some cells (Figure 2).

Following upon the secondary phase of convective initiation, the line of cells starts accelerating eastward, with the reflectivity

signals exceeding 45 dBz at $z = 3$ km propagating to $x = 10$ km after 75 minutes and almost $x = 15$ km after 100 minutes

(ENS-03). The line becomes increasingly inhomogeneous in the y-direction and exceeds 45 dBz at $z = 3$ km (about a km

below melting level) over a longitudinal stretch of 15-20 km during this period, with very intense cores of 65 dBz. The latter is

very understandable given the convective environment by the low level cold pool in combination with about 2000 J/kg CAPE

(see Weisman and Klemp, 1982) and 14-15 m/s of wind shear over the lower 2.5 km. It leads to excellent conditions for very

intense linear convection.



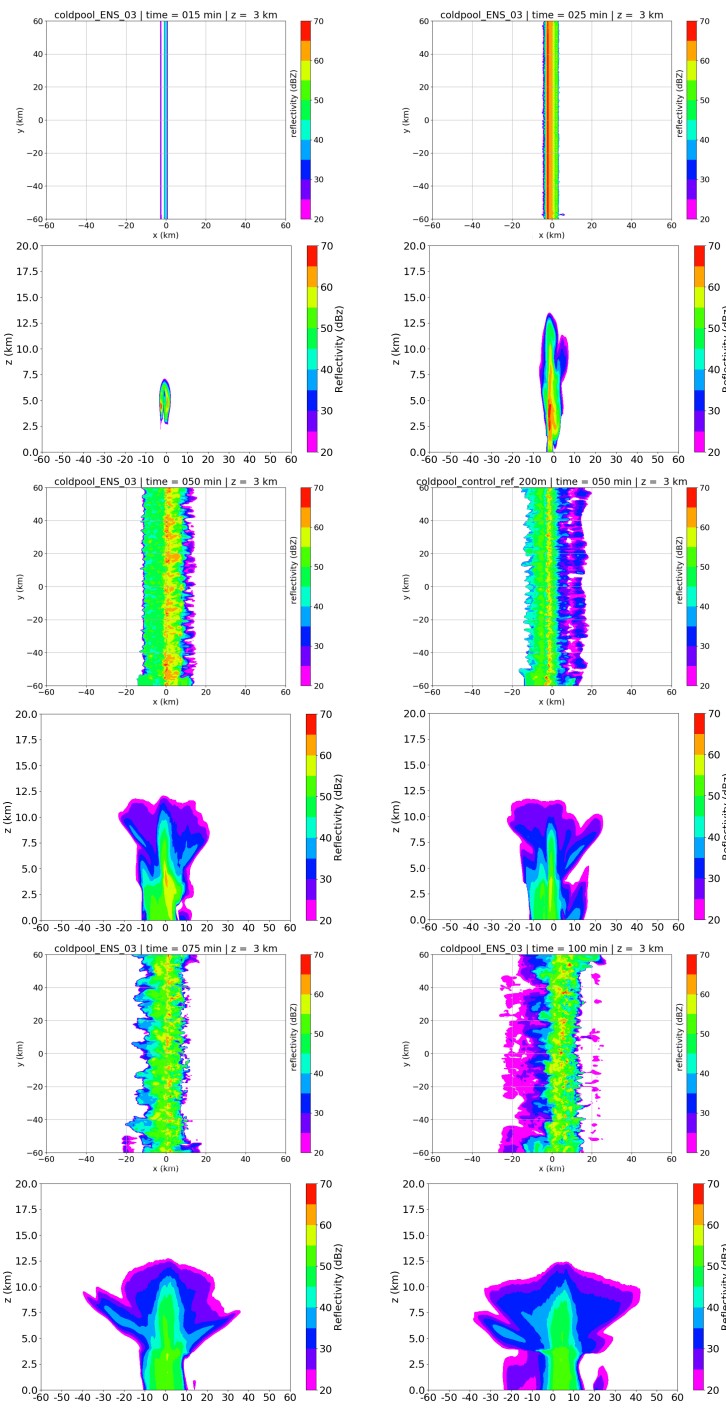

**Figure 2.** Evolution of simulated reflectivity for ensemble member 3 and the reference simulation (only $t = 50$ minutes) at $z = 3$ km. A horizontal cross section is followed by corresponding x-z cross section directly below, with median reflectivity along the squall line. Left top (a,c): $t = 15$ minutes, right top (b,d): $t = 25$ minutes, left center (e,g): $t = 50$ minutes (ensemble member 3), right center (f,h) $t = 50$ minutes (reference simulation), left bottom (i,k): $t = 75$ minutes, right bottom (j,l): $t = 100$ minutes.



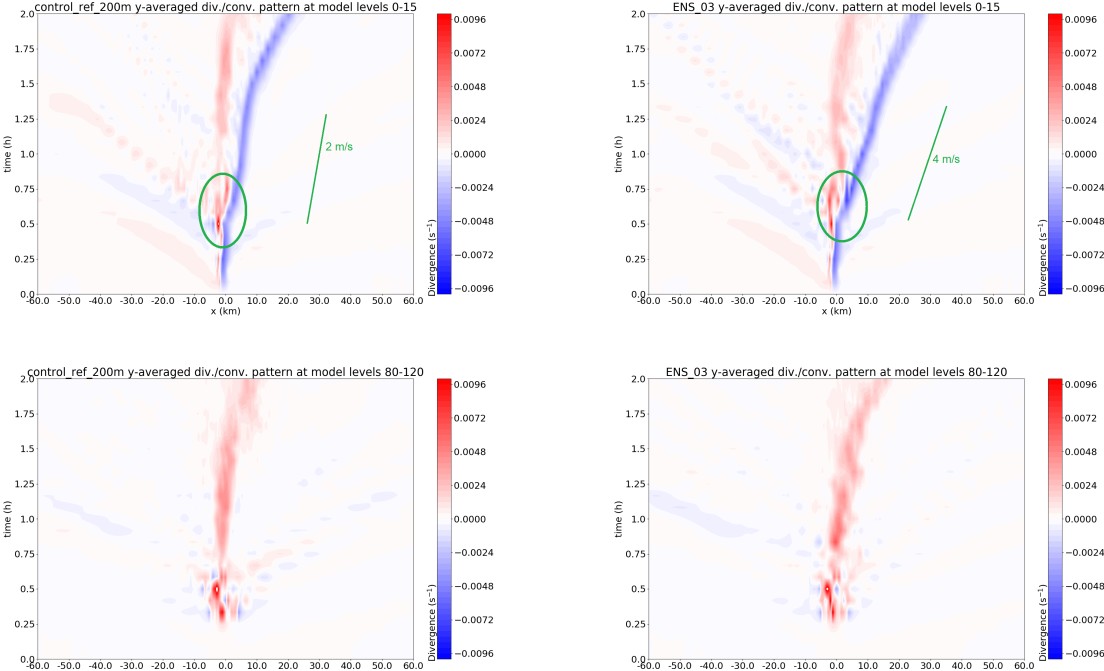

**Figure 3.** Space-time distribution of low level (0-1.5 km; top) and upper tropospheric (8-12 km; bottom) convergence features in the reference simulation (left) and ensemble member 3 (right), averaged over all $y$ and given $z$. A stage of particular interest that is analysed in the text is highlighted with a green oval, as well as the slope of the convergence zones halfway the simulation.

## 3.2 Detailed comparison of two example simulations

The upcoming section focuses on the two illustrative simulations, before moving on to the intraensemble variability in Section
3.3. Focus lies on comparing the reference simulation and ensemble member 3 (see also Figure 2), because they often appear at opposite ends of the distributions in most diagnostics. The evolution of convergence and divergence zones in space and time is described (Section 3.2.1). Describing their motion is needed to understand how the sources and sinks of convective updrafts displace with time.

For further analysis of the contrasts between the simulation pair, these two simulations have been selected and repeated with
passive tracers, as described in Section 3.2.2.

### 3.2.1 Convergence and divergence zones

Figure 3 shows a set of Hovmöller-like diagrams of divergence (red colors) and convergence (blue colors) in the lower and upper troposphere. The top panels demonstrate that the low level convergence zone accelerates eastward in both simulations





after a nearly stagnant position in the first 30 minutes or so, in agreement with displacements of convective cells in the previous section (Section 3.1). After this stagnant stage, several patterns indicating eastward acceleration of convergence and divergence zones are visible. First, in the reference simulation the convergence zone quickly accelerates to stationary eastward propagation of about 2.5 m/s in the following 45-50 minutes, whereas it does to about 4.3 m/s in ensemble member 3. Second, another stage of acceleration of this zone occurs at the end of the simulations, but this is after the main time frame analysed. Third, a weak

acceleration of the divergence zone in the wake of the former zone occurs, most prominently in ENS-03.

In addition to these main low level convergence features, convergence and divergence patterns associated with gravity waves are also visible in Figure 3. These comparatively weak waves lose amplitude in time. Apparent intermittent behavior in both figures for the western half of the domain is probably caused by the output interval of 5 minutes, approximately the apparent pulse frequency of these features. The bottom panels of Figure 3 reveal a nearly stagnant and slightly diffuse patch of upper

tropospheric divergence at $x \approx 0$ km, collocated with the squall line core. Propagation of this upper tropospheric divergence is clearly restricted to the second simulation hour for ensemble member 3. Furthermore, the consequences of developing convective cells (Section 3.1) are visible only after the first 15 minutes. Lastly, as in the lower troposphere, one can also see patterns associated with comparatively weak gravity waves propagating in the upper troposphere, which leave the model domain after the first hour.

As opposed to the reference simulation a sudden displacement of the low level convergence zone happens after 0.6 hours (green oval, Figure 3), with a double divergence zone in its wake in ensemble member 3 and an increase in amplitude of the convergence directly after. In the reference simulation this event happens in a smoother way: without a strong increase in amplitude of the convergence. On the contrary, the convergence zone also jumps by about a couple of km in ENS-03 and does not in the reference simulation. The jumpy displacement and amplitude increase of the convergence zone in ENS-03 is related

to the initiation of secondary cells ahead of the squall line core (Section 3.1).

### 3.2.2 Secondary convective initiation and tracers

Tracers are a useful tool to identify the pathway that inflow follows: it provides quantitative information on the destination of specific inflow layers as part of the air originating in these layers moves through the convective cells that the squall line consists of. Of course the tracer analysis is restricted to the identification of transport from time of installation until given output time

and to a coherent air mass. Tracers PT1 and PT2 are initiated below altitudes of 2.5 km ahead of the squall line, so the upward transport of mid level entrainment and its effect on the magnitude of convective overturn cannot be investigated. To evaluate the contrast between the simulation pair (Section 3.2.1) in more detail, passive tracers have been introduced in the inflow of the reference simulation and ENS-03. Tracer PT1 was implemented below 800 m between $x = 0$ and $x = 30$ km at $t = 0$ and PT2 was simultaneously initiated over the same horizontal region at all model levels where $1600 < z < 2400$ m. All concentrations

were initially set to 0.001 kg/kg.

When one looks at the passive tracer concentration difference between ENS-03 and the reference of PT2 after 30 and 35 minutes is examined, peculiar patterns are found (Figure 4). The pattern consists of a surplus in PT2 after 30 minutes at the location of the black "X" in ENS-03 and a surplus about 1 km lower in the reference simulation, amongst others.







**Figure 4.** PT2 concentrations difference between reference and ENS-03 after 30 min (top, a), 25 min (left bottom, b) and 35 min (right bottom, c). Red indicates a tracer surplus in reference simulation, blue in ENS-03. The black in a X marks the gravity wave crest in which vertical displacement leads to triggering a new line of convective cells in the squall line, as we will see later. Note that only half of the 120 km domain is shown.

More precisely, a dipole structure in the PT2 surplus from $z = 4$ km to $z = 6$ km around $x = 0$ km at $t = 25$ minutes suggests
a slight shift between the developing convective updrafts between the simulation pair in Figure 4. At $t = 30$ minutes the
dipole has elongated vertically: the updraft core is shifted eastward by a few grid cells in ENS-03 compared to the reference.



Furthermore the PT2 surplus at the location "X" in ENS-03 is very notable. At this location PT2 concentrations are much higher in ENS-03 than in the reference simulation at $z = 3$ km, a few km east of the x-axis. About a kilometer below PT2 concentrations are lower at the same time in ENS-03, at the lower flank of the source layer (Figure 4). A difference of opposite

sign occurs at the top of (red; and below: blue) the PT2 source layer some 10 km to the east of X. After 35 minutes, the same signal has propagated eastward by some 5-7 km and the blue signal at location X has extended upward, but the red has not. Consequently, PT2 transport is massively different between ensemble member 3 and the reference simulation, from t = 30 to t = 45 minutes: only 7.0% of the PT2 mass moves from its low level source layer to levels above 4 km height in these 15 minutes in the reference simulation, whereas 17.6% does in ensemble member 3. Furthermore, 11.7% (ENS-03) is flowing to the UT

($z > 6$ km) versus 6.0% (reference simulation). This implies a very large difference in convective tracer overturn and hence convective mass fluxes will be significantly larger in ensemble member 3 than in the reference simulation.

Figure 5 displays the PT1 distribution after 55 and 75 minutes. Both PT1 and PT2 (not shown, similar patterns) indicate that

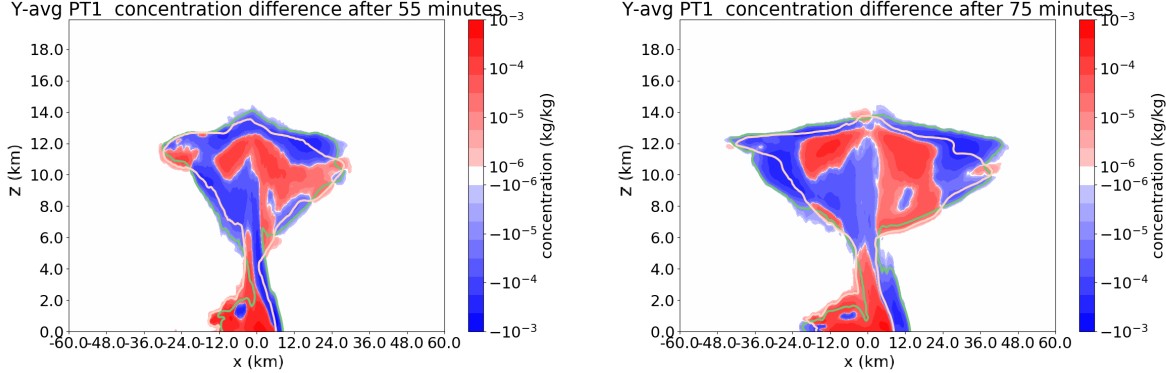

**Figure 5.** Difference in PT1 concentration between ENS-03 (blue: higher) and reference (red: higher), after 55 minutes (left) and 75 minutes (right) in colors. In addition, the $q_{PT2} = 1e-5$ kg/kg isoline is shown for both simulations (salmon color: reference; green: ENS-03). Y-average is now taken over the limited subspace, consisting of 40 km$> y > -20$ km.

the top of the region with lower tropospheric tracers is lower in the reference simulation compared to the ENS-03. As the passive tracers are initiated at the location of source air for the convection, they reveal the source region of convected mass.

Two additional patterns are visible in the aforementioned figure: first the near-surface eastern boundary of PT1 moves faster eastward (in blue, ENS-03) than in the reference simulation (red). Second, the tracer is laterally spread out further in ENS-03 (green contour, 5b) than in the reference simulation.

The three aforementioned patterns that PT1 shows (Figure 5) are a consequence of differential mass transport within the convection. Together with the patterns seen in PT2 earlier in this section, increased convective overturn in ENS-03 may lead to

all of the three patterns. Extra mass overturn is consistent with the reflectivity patterns from Section 3.1.

The upper tropospheric divergence as visually demonstrated by the passive tracers is strongly reduced in the reference (pink) after 75 minutes. The computed upper tropospheric mass divergence up to 75 minutes turns out to to differ by no less than 38.3

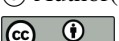



(!) %. Notably, the precipitation flux over this interval is not as strongly affected: net latent heating left behind is only 23.5% higher in ENS-03 than in the reference simulation.

### 3.2.3  Vertical velocity at "X" and interpretation

Location "X" in Figure 4 has drawn specific attention, with a contrast between upward and downward displacements of PT2 in the reference simulation and ENS-03. These downward / upward displacements in the trough / crest are resembling gravity wave patterns, as it has been identified in the previous section. Furthermore, consequent differential vertical overturn of mass has been identified with tracer evolution directly afterwards.

The patterns identified in the former section therefore likely have a strong connection with vertical velocities at location "X", which could be affected by both variability in the gravity waves passing by first and then possibly by consequent variability in the potential for convective initation, corresponding to the lifting of air above its LFC.The mean $w$ along the squall line at location X ($t = 30$ min) is +0.35 m/s in ENS-03 and -2.76 m/s in the reference, with along-line deviations of up to 1.5 m/s about the mean. Therefore, a large portion of X along the length of the squall line consists of favorable locations for initiation of convection in ENS-03 with $w > 0$, but not in reference where $w$ is well below 0. Other ensemble members are at intermediate values between those two opposites, with a mean of -0.93 m/s. Given the spread of about 1.5 m/s in local $w$ about the mean, most members contain some limited areas favorable for the secondary convective initiation.

### 3.3  Ensemble squall line variability

In this section the focus of the analyis is shifted towards the full ensemble. The former two simulations at both ends of the ensemble have already illustrated some important signals of squall line variability. Covariability between source and target variables within the ensemble will be investigated in the following section and then robustness of the signal is also assessed. Here, the source variable refers to a variable that works as an independent variable in the problem. The signal in a source variable precedes that in the target variable, acting as dependent variable in the analysis, with a mostly positive time lag in between. Both contain information about the squall line relative flow in the context of the upcoming analysis in Section 3.3.2. Source variables are the vertical velocity at X (Figure 4) (Sections 3.3.2, 3.3.3) and the acceleration of the cold pool (Sections 3.3.1, 3.3.3), as in described in Section 3.2. The role of source variables will be interpreted in the Synthesis (Section 4).

### 3.3.1  Location of the cold pool edge

The cold pool location and its time derivative, propagation $v_{cp}$, is computed at each output step by taking the $\partial_x$ of $\rho$ at the lowest model level and detecting its maximum over the x-direction. This computation over the zonal direction is done for each coordinate in the $y$-direction. The cold pool position and velocity are obtained by taking the average value of the resulting $x$ over $y$ and converting that $x$ to a grid cell index. The result of the aforementioned procedure is used as mean cold pool edge location and from now on defined as cold pool edge for a given ensemble member and time.

While some of the further calculations in this study are done in the domain reference frame, other calculations in the remainder



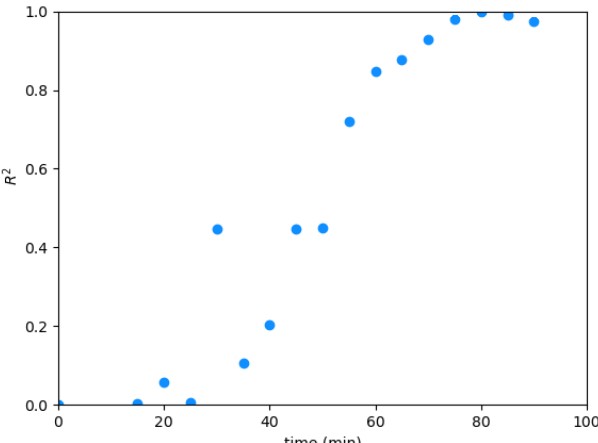

**Figure 6.** $R^2$ of auto-correlation as a function of time for the cold pool position among 10 ensemble members, with the position at $t = 80$ minutes as reference correlation of 1. The points at $t = 5$ and 10 minutes have been ommited, as there is no variability in cold pool edge location among the ensemble members (yet).

are performed relative to the detected cold pool edge. Calculations relative to the cold pool edge will be explicitly denoted as
such.

With this definition $v_{cp}$ differs by a factor of 1.7 within the ensemble over the interval 30-75 minutes for the reference and ENS-03: 2.5 vs. 4.3 m/s. That simulation pair only differs by 47 m in initial interface height of the shear layer ($< 2$ %)! These simulations are on the outer ends of the ensemble distribution of cold pool velocity.

In this section we continue with the analysis of the whole ensemble distribution. The average displacement of the cold pool
edge at the surface has been evaluated and auto-correlated with its own location at $t = 80$ minutes. On purpose this is also directly after the end time of the subinterval that is focused on in much of this manuscript (Section 2.3).

We cannot calculate very robust values for the auto-correlation coefficient, given the limited ensemble size of $n = 10$. On top of the limited ensemble size, the limited spread in cold pool edge location is initially another substantial source of uncertainty in the auto-correlation coefficient of the cold pool propagation. After only 25 minutes the spread is restricted to 1 grid cell. As
the ensemble spread in the cold pool location increases to 4 grid cells after 40 minutes and keeps on increasing linearly beyond, the second substantial source of uncertainty is reduced and then eliminated. That means with some care, one can interpret its time evolution and use it to increase the insight in the squall line evolution within the ensemble.

As one can see from Figure 6, the auto-correlation function shows an S-like shape. Since the instantaneous values of the function have to be read carefully (previous paragraph), roughly three stages can only be distinguished. First, there is a stage
prior to establishment of the intra-ensemble variability in cold pool location. Second, there is a stage where the intra-ensemble variability of cold pool location is establishing, corresponding with the growth stage of S-curve. Third, there is a stage where the intra-ensemble variability is established.





The curve shows the stage prior to establishment lasts until about $t = 30$ minutes. After this pre-established stage, the growth rate is maximized around 40-45 minutes. Therefore the growth stage lasts from about 35-50 minutes. The growth stage is

the stage where the variability in cold pool location is established. Lastly, from about 50-55 minutes onward the cold pool location variability within the ensemble has been established and is maintained until 80 minutes, where $r = 1$, and beyond. In other words, cold pool locations relative to one another are not varying substantially after 55 until 100 minutes. The ensemble variability structure is maintained, or equivalently, the established $v_{cp}$ is roughly fixed. Both the standard deviation and the difference between maximum and minimum $x$ of the cold pool edge increase nearly linearly beyond $t = 45$ minutes (not

shown). This was also visually apparent for ENS-03 and the reference in Figure 3.

### 3.3.2 Ensemble sensitivity analysis

The ensemble sensitivity technique is applied to assess statistical patterns that exist in the squall line ensemble. Basically the ensemble sensitivity finds the regression line through the dimension of the ensemble members that describes the best fit between two variables $x_1$ and $x_2$, like in any analysis of covariance patterns (see also Hanley et al., 2013; Bednarczyk and

Ancell, 2015). Bednarczyk and Ancell (2015) discusses some limitations of this analysis for a convective case, which apply specifically if the (spatial) extent of the source variable is not exactly known and a proxy has to be used as in their case: deep convection was parameterized and precipitation or reflectivity related output had to be used instead. On the contrary, in Section 3.2.2 and Figure 4 the spatial extent of a variable of interest is clearly delineated for this case and it also directs at the source variable, local $w$.

Since squall line variability is investigated here, y-averaged variation in $u$ in $x$-$z$ plane is of particular interest, especially within the squall line core. This core area with the updrafts, downdrafts and some inner portion of the anvil is the target variable of the ensemble sensitivity analysis. Tailored statistical testing will be used to investigate the robustness of the most important signals.

Now we will continue with the ensemble dataset and explore the sensitivity of squall line circulation to a precursor pattern in

the velocity field. Therefore an ensemble sensitivity analysis targeted at $u$ in relation to $w$ at location X in Figure 4 is done. This $w$ at X is averaged along the length of the squall line over five by five grid cells in the $x$-$z$ plane. A covariance analysis of the latter variable is done with the y-averaged $u$ in the x-z plane at any t in the simulation as covariant. The obtained ensemble sensitivity of that covariant is chronologically discussed in this section. Note that the ensemble sensitivity analysis is carried out in a cold pool edge relative framework: as this cold pool edge moves by up to 30 km during the simulations, correction

for its displacement in $x$ overlays various x-coordinates and thereby shrinks the available domain for the ensemble sensitivity analysis in the zonal direction by approximately 30 km from 120 to about 90 km.

Initially, during the first 15 minutes of simulation time, the signals revealed by the ensemble sensitivity analysis contain very low amplitude patterns, that are virtually unnoticeable in terms of u-velocities associated with it. Based on their geometry

and fast propagation speed, these waves have to be low frequency sound waves with wave lengths on the order of 5 km and can spontaneously develop from density anomalies. As the acoustic waves are from a practical perspective unrelated to the





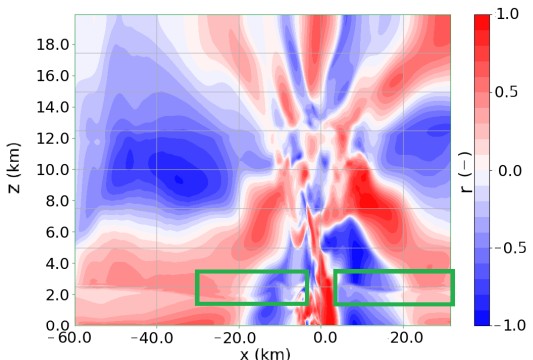 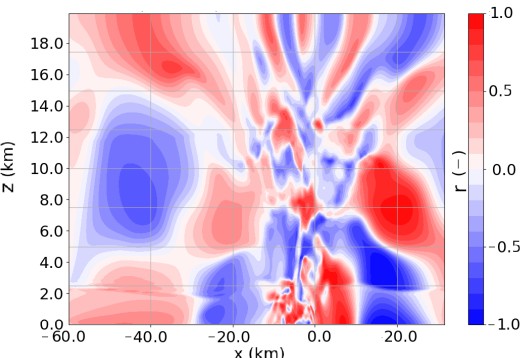

**Figure 7.** Correlation structure obtained from the ensemble sensitivity analysis. Left: $t = 30$ minutes, right: $t = 35$ minutes. On the left green rectangles are illustrating where "undulating wave signals" (see text) that resemble signal of gravity wave activity are located. One can see a gravity wave crest at $z_i \approx 2500$ m, propagating from $x = 4$ km to $x = 9$ km. With positive vertical velocity perturbations and inbound/converging $u$ winds upon this crest, enhanced forcing for convective initiation ($x = 4$ km, $z = 3$ km, $t = 30$ minutes, X in Figure 4) is correlated with our target variable.

convection, their relation with the squall line developments can be ignored.

After 15-20 minutes, the area within and in the immediate surroundings of the convective bubble, along the y-axis, shows some covariability with the source variable. Initially this signal is restricted to some of the middle troposphere and the immediate wake of the cold pool edge. The magnitude of the covariability is initially small, below 0.1 m/s, but very locally reaches 0.5-1 m/s by 20 minutes.

The somewhat turbulent signal has propagated vertically after 25 minutes and extends from the surface up to about $z = 12.5km$, which corresponds with the region of developing cells. Its amplitude is now also noticeable in terms u-values: in the whole troposphere about 0.5 m/s. At this same instance, one see the first association of $w$ at X with an undulating wave signal travelling at the interface level $z_i$, as exposed by a trough and air that has sunk. That signal resembles a gravity wave, consistently with signals from the tracer analysis. None of the signals in the first 20-25 minutes pass the statistical robustness threshold: the amplitude may be considerable, but they are not systematically associated in a linear way with the source variable of the analysis.

At the next time step, the former trough has propagated westward and a crest develops in the immediate wake of the cold pool edge ($x = -10$ km), while ahead of the cold pool edge a first signal of gravity wave activity is apparent ($x = 5$ to $x = 15$ km; Figure 7). Through $w$ the latter wave signal is associated with the strong signal of convective initiation in ENS-03. On the contrary, this secondary phase of convective initiation does apparently not occur in the reference simulation. In the following three time steps many more faster gravity wave signals propagate away from the source region. This source region of warm air upstream of the squall line would definitely pass the statistical significance test as a robust signal for $t = 30$ until $t = 60$ minutes. As the circulation is effectively almost 2D that is partly inherited by the fact that the circulation in the x-z plane





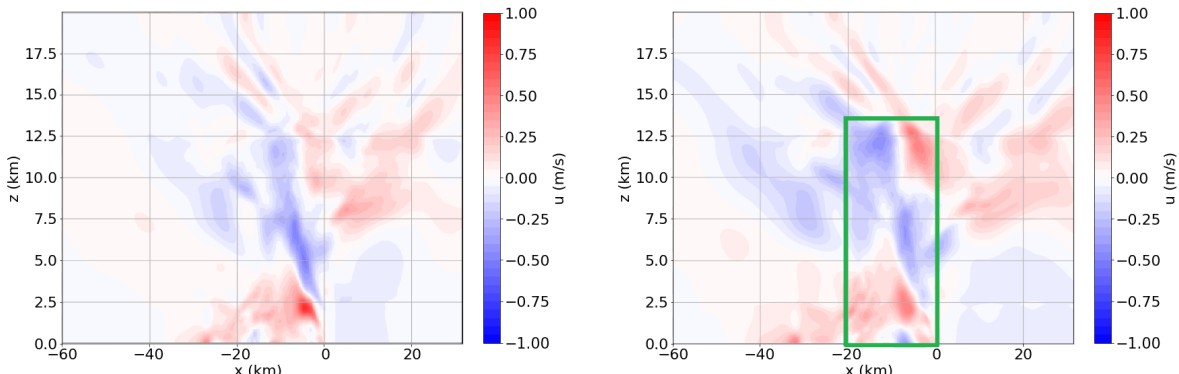

**Figure 8.** Squall line structure with associated y-averaged $u$ (relative to the cold pool edge) after (left) $t = 50$ minutes and (right) $t = 55$ minutes. Among the main features are enhanced convergence in $u$ relative to the cold pool edge around $z = 2.5$ km and $x = -4$ km, as well as enhanced divergence from $z = 7$ to $z = 13$ km. On the right, one can see extended divergence patterns (slightly) above $z = 13$ km, which is likely associated with an overshooting cloud top with divergent outflow in $u$. The central area of squall line circulation (see text) used for significance testing of the identified squall line circulation anomaly is marked with a green rectangle (only right figure).

contains the source and target variables of the statistical analysis: $w$ at t = 30 minutes and $u$.

Following this stage of apparent gravity wave activity, a new stage in the evolution begins. To investigate this stage the focus is shifted from the correlations of the ensemble sensitivty analysis to the second covariant, namely the horizontal circulation perpendicular to the squall line ($u$).The signals in $u$ move from upright multi-wavenumber in the vertical, centered at the con-

vective cells to one that is aligned with the tilted cold pool and another contribution that causes upper tropospheric divergence away from the convective cells in the anvil. Both of these signals of variability are associated with the squall line circulation, based on the location, extent and orientation and occur after 40-50 minutes. Synchronously, the associated $u$ variability also exceeds 1 m/s, maximizing at altitudes of 1.5 to 8 km (Figure 8a). The features of zonal flow still amplify in the higher regions of the troposphere and even up to the region of convective overshoot (Figure 8b), but also gradually propagate away from their

source region in the following 30-40 minutes.

The signal of the previous paragraph is statistically very robust: in the central 20 km of the squall line and at altitudes up to the tropopause a fractional area $f$ of 0.56 passes the statistical significance test at $p = 0.05$ and $f = 0.35$ at $p = 0.0125$ (Figure 8b). This is more than an order of magnitude larger than expected values (see also Wilks, 2016).

After 80 minutes, the u-variability associated with $w$ at X in figure 4 has weakened. Later some renewed variability occurs at

$t = 90$ minutes, behaving similar to that in the previous paragraph: propagating away from its initiation region slowly while dying out. The initiation region has moved slightly upstream and slightly higher up in the troposphere. In addition, the signals that likely occur due to gravity wave activity around and below $z_i$ remain to be associated with our source variable, but that signal is not robust.

The circulation revealed by the ensemble sensitivity analysis demonstrates anomalous convergence of mass at $z = 2 - 3$ km,





rearward of the cold pool edge (Figure 8). The convergence signal consists of enhanced easterly flow in the updraft region which gradually rises and moves upstream relative to the cold pool, due to the updraft tilt. The feature with easterly flow anomalies moves upward, extending initially ($t = 45$ minutes) from the surface to 8-9 km altitude. On its rearward side a westerly flow anomaly does roughly the same, but sticks to lower levels and dissipates earlier.

At the same time, another main feature occurs in the upper troposphere in the sensitivity signal: an enhanced upper tropospheric

divergence at $t = 50$ minutes, which overshoots into the first 1-2 km of stratosphere at $t = 55$ minutes (Figure 8). This upper tropospheric pattern itself diverges in time and fades outward after about $t = 80$ minutes. Some patterns with reversed anomaly seemingly act as compensation appear from $t = 90$ to $t = 115$ minutes, but happen outside of the focal time frame of this study. Furthermore, other diagnostics have not revealed any patterns that would imply compensating behavior.

### 3.3.3 Downdraft variability and cloud tops

In this section the downdraft characteristics are discussed in more detail. Furtheremore the cloud tops are evaluated and then the statistical relations between these diagnostic variables and the two precursors are quantified, namely $w$ at X in Figure 4 and cold pool acceleration to $v_{cp}$.

**Downdraft detection**

To compute the downdraft mass flux, area profiles of downdrafts and conditional vertical velocities in downdrafts, first all

grid cells with negative (positive; for updraft detection) vertical velocities ($w < 0$ m/s) and a minimal total cloud hydrometeor density of 1e-4 kg/m$^3$ were selected. The downdraft area is given by the number of grid cells occupied in each layer. This computation method is inspired by and adjusted from Varble et al. (2020). Subsequently, gravity wave contributions from saturated parcels are removed, by removing any grid cells where the total density ($\rho_{moist\_air} + \rho_{water}$) was lower than its horizontal average over that model layer. Without the latter correction, downdrafts are highly overestimated in the near-tropopause region,

where gravity wave activity is leading to a lot vertical transports with cloud/hydrometeor content that is subsequently undone by the reverse motion (see also Pandya and Durran, 1996).

Any grid cell classified as downdraft is then used to calculate relative area at height $z$ and conditional mean of the vertical velocity $w$.

**Cloud top detection**

Cloud top detections have been tested using various thresholds. In all of these, either the simulated fraction of reflectivity beyond 15 dBz or an ice fraction above 1e-6 kg/kg were used as definition of a cloud detection mask. These are two very different masks and leads to cloud top distributions separated by almost 2 km in the vertical. Subsequently, the relative area covered by the cloud mask at each z was computed for $t = 60$ until $t = 90$ minutes. The interval was chosen, because the cloud tops start to slightly overshoot locally after 55-60 minutes and the anvil spreads out around a relatively flat equilibrium level (see

also Figure 5). The cloud top was defined to lay between the uppermost layer with a 0.1 cloud fraction and the following layer where the cloud fraction sinks below 0.1. Fractions of 0.3 and 0.01 were also tested, and are very closely related to the cloud tops at fraction 0.1. The $fraction = 0.1$ visually appears to represent the typical location of cloud tops best. The reflectivity threshold leads to slightly increased spread between the ensemble members compared to cloud ice and was selected in favor of





the cloud ice threshold.

First, the y-averaged downdraft and updraft fluxes are displayed in an x-z cross section for one random ensemble member in

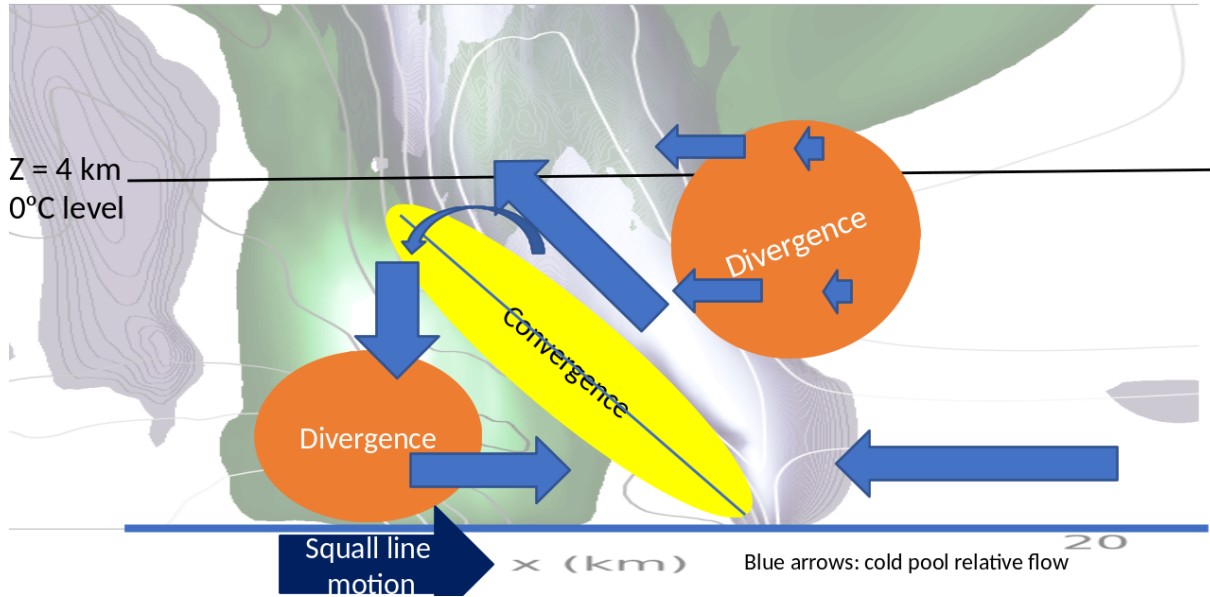

**Figure 9.** Schematic picture of the cold pool circulation in x-z cross section: green coloring shows updraft mass fluxes, purple coloring shows downdraft mass fluxes. The white isolines indicate easterly flow (-9 m/s, every 3 m/s), while darkest grey ones indicate westerly flow (+6 m/s). One can identify convergence and divergence zones from these contours, which are conceptualized with yellow and orange ovals. The cold pool relative flow is given by blue arrows. Note that the classical pattern of a rear inflow jet (Houze, 2004, and references herein) is practically absent here and that the upper (roughly) half of the troposphere is omitted. The squall line propagation is given by the dark blue arrow at the bottom.


Figure 9. Most of the downdraft variability within the ensemble is achieved through downdraft area. This is shown in Figure 10, but this figure shows instantaneous values, as opposed to values used for statistical diagnoses. The mean conditional downward velocities are less strongly varying than downdraft area. The time averaged maximum area covered by downdrafts in the lower kilometers varies by around 20% throughout the ensemble, while maximum mass fluxes even vary by up to 40% among the

members. Its maximum variation is reached at low levels. As the downdraft area at the mean level of maximum downdraft flux varies by up to 27%, it explains most of the downdraft variability. However, additional contributions come from the conditional mean of the downdraft velocity and perhaps to a very small extent from the density.

In Section 3.3.2 a tilted convergence feature was identified, which is amplified with increased $w$ at location X in Figure 4. A portion of the extra strong easterly flow in the updraft regions halts in the convergence zone and updraft air on the upstream

side leaves the updraft region for the convergence zone at the upstream tilted cold pool edge. Converging with nearly stagnant



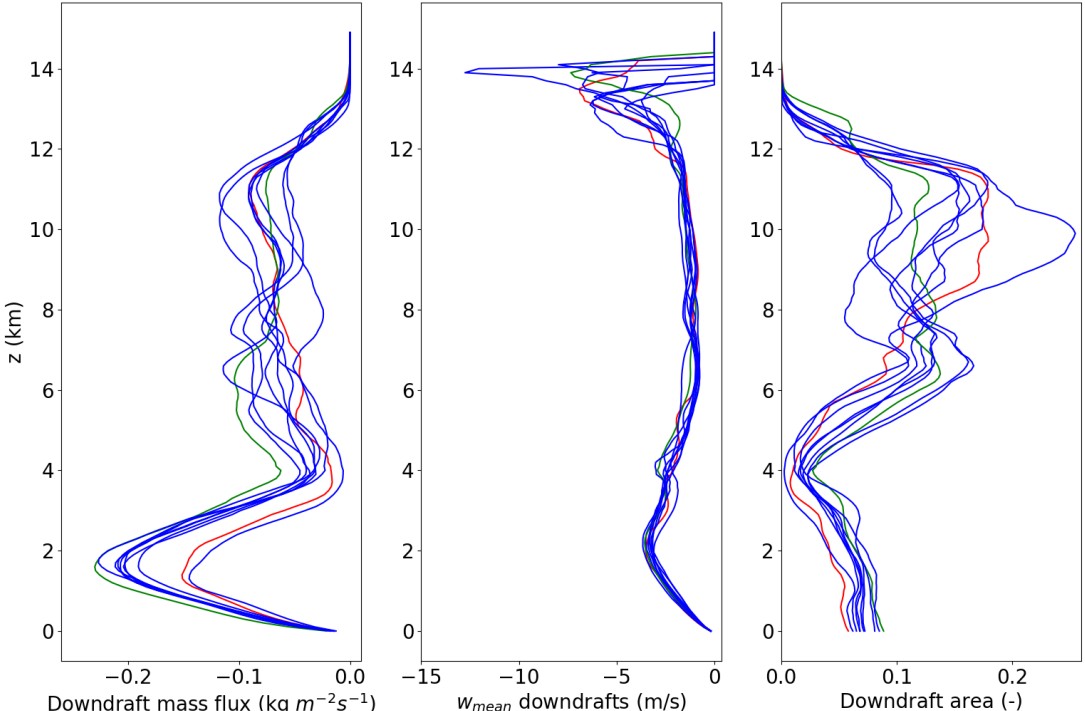

**Figure 10.** Instantaneous profile of squall line downdrafts at $t = 75$ minutes. Left: downdraft mass flux, center: mean conditional $w$, right: fractional area. Red: reference simulation; green: ENS-03; blue: other ensemble members.

air or weak westerly flow that moves approximately at $v_{cp}$), the downdraft fluxes below melting level are amplified at $z = 1 - 4$ km (Figure 10). The circulation described in the earlier part of this paragraph could be thought of as a subcirculation within the squall line (Figure 9): a cold pool circulation. Positive covariance of the updraft and downdraft variability within the ensemble would be expected, based on the presence of the convergence anomaly in Figure 8. That covariance therefore also implies a potential response of downdraft mass flux to the secondary convective initiation in Sections 3.1 and 3.2.

The statistical links between mean $v_{cp}$, $w_{t=30}$, downdraft characteristics and other variables that have been diagnosed in detail are presented in Table 1. Significant correlations between the downdraft properties and vertical velocity at X after 30 minutes are clear. The latter variable affects convective initiation, relates to overturned mass and in the end to the squall line circulation. In Section 4 these and other connections will be reviewed in more detail.





**Table 1.** Statistical relations between quantities of interest. $R^2 > 0.4$ is significant at $\alpha = 0.05$. $Z_{max}$ in the table refers to the level where the low level downdraft flux maximizes ($z < 4$ km).

| Precursor, Possible driver | Effect, target | Time interval precursor (min) | Time interval target (min) | $\mathbf{R^2}$ |
|---|---|---|---|---|
| $w$ at X | $v_{cp}$ | 30 | 30-75 | 0.612 |
| $v_{cp}$ | Downdraft area at $z_{max}$ | 30-75 | 30-75 | 0.611 |
| $v_{cp}$ | Downdraft flux at $z_{max}$ | 30-75 | 30-75 | 0.799 |
| $v_{cp}$ | Precipitation flux | 30-75 | 0-75 | 0.927 |
| $v_{cp}$ | Upper tropospheric divergence | 30-75 | 0-75 | 0.595 |
| $v_{cp}$ | Cloud top height | 30-75 | 60-90 | 0.559 |

## 3.4 Error growth in the squall line simulations

To conclude the results with, this section is about error growth in the squall line simulations.

The error growth will be investigated through the ensemble evolution of the statistical spread in the winds with and without a correction for the cold pool propagation. In other words, a difference between an Eulerian and feature relative perspective is identified. By illustrating the contrasts between the two ways of looking at error (or equivalently variance, spread, but the term error will be used in this Section) growth an attempt is made to deduce some idea of the processes contributing to the error during the main development stage of ensemble errors (about 25 to 60 minutes).

A limitation of this method is that only net growth of errors can be detected. Some processes can destroy errors (e.g. diffusion) and so the method provides additional insight only when it finds growth in the error: a process leading to highly unbalanced tendencies dominates error tendencies. On itself, it has limited abilities to describe what is going on, but the combination with other diagnostics makes it a useful tool to learn about the relevant processes. Negligible initial errors typically lead to error growth to a certain climatological envelope on long time scales and infinite time limit, which makes the error growth analysis a useful tool here.

Just like the ensemble sensitivity, diagnosis of error growth is targeted at variability in squall line circulation here and therefore applied to the $u$ winds in the $x$-$z$ plane. On the contrary, a more complete energy metric is usually considered in error growth studies (Selz, 2019; Zhang et al., 2007; Zhang, 2005), which includes $u^2$, $v^2$ and may sometimes also include $T$. The growth of ensemble error as a measure of ensemble variability is mapped by the difference winds between ensemble pairs. The simulations that such a pair consists of are usually assumed to fulfil a perfect "error" assumption, analogous to a perfect model assumption (see Selz, 2019): only the error needs to fulfil this assumption and not the full model itself. With this, growth rates of error energy (difference kinetic and/or difference total energy, Zhang (2005)) can be compared with expected behavior. In this study, this metric will be used to compare the growth of variation in circulation revealed by $(\Delta u)^2$ between cold pool relative and grid point framework, as well as to define in which stages of the simulation the errors growth in specific ways and faster than in other stages. Zonal wind is averaged over y, wherever the cold pool relative framework is used.



Figure 11 shows the error growth curves, where differences in $u$ winds between each ensemble member and the reference run are treated as "errors". Note that the $v$-component is not included, but it is typically at least comparatively much smaller than $u$,

and $u$ should therefore reveal the dominant error patterns. Nonetheless, the part of the domain downstream of the squall line is not fully included. This is because about three quarter of the domain can be stacked on top of each other only after one corrects for the cold pool location (see Section 3.3), as the cold pool edge progresses in time over up to a fourth of the domain.

A first major stage of growth occurs between $t = 20$ and $t = 30$ minutes in both the corrected and uncorrected $(\Delta u)^2$. This stage in error growth is likely driven by decorrelation of gravity wave phase and amplitude and to some extent by differences

the convective initiation, as has been suggested in Section 3.2.2 for example. This early stage should not cause much variation between the corrected and uncorrected error curves, because the cold pool edge does practically not move yet. However, some variation can occur due to the inclusion of the full domain for the grid point based comparison versus only three quarter of it for the $v_{cp}$ corrected and y-averaged curves. In practice, differential error growth between the corrected and uncorrected curves is seen after 25-30 minutes.

Another second major stage of error growth occurs from 40 or 45 minutes to 60-65 minutes. The variability in squall line flow (Figure 8) strongly develops in this stage. In this stage the slope between the corrected and uncorrected curves is systematically different, with much larger slopes for the grid point based curves. In other words, a substantial portion of variability in squall line relative circulation can be explained and is probably induced by $v_{cp}$ variability. However, even if we subtract the cold pool displacement, some of the ensemble members still demonstrate strong error growth, whereas others do hardly show error

growth compared to the reference simulation. The impact of non-linearities and feedbacks that affect error growth as well are clearly present to some extent and not negligible, but during this stage non-linearities definitely do not dominate over $v_{cp}$ effects. In other words, these non-linearities do not dominate the error growth in the ensemble. With a very rough estimate one could say that after about 60 minutes of simulation time about a third to half of the error is related to differential cold pool propagation and another third to half of the total errors was pre-existing after 30 minutes. Even though the latter two

contributions are not fully independent, this means that non-linearities must be of a magnitude that is somewhat comparable to the two.

Moreover, both the corrected and uncorrected error curves seem to grow linearly in the second stage. In combination with the relatively constant $v_{cp}$ values in each simulation (Section 3.3.1), it implies that the cold pool acceleration linearly explains a substantial proportion of variability in the squall line circulation and possibly even the majority of the errors.

Another notable feature in between the two identified main stages of error growth in the error growth curves is the relative flatness of $v_{cp}$-corrected curves from 25-30 to 40-45 minutes. This must be explained by a combination of regions of error growth and other regions where errors decay. Error growth must occur in the region where the secondary phase of convective initiation happens, but may also happen in regions elsewhere. Spatial distributions of squall line relative difference winds strongly suggest that the decay of errors mostly happens at locations of difference wind maxima after 25 minutes in the upper

tropospheric region with cell cores, around $x = 0$ km (not shown). That implies the relatively flat curve after 30-40 minutes is not inconsistent with local error growth due to secondary convective initiation.

After the first 60-65 minutes (Figure 11), the cold pool corrected curves do not seem to grow anymore, but errors remain rather





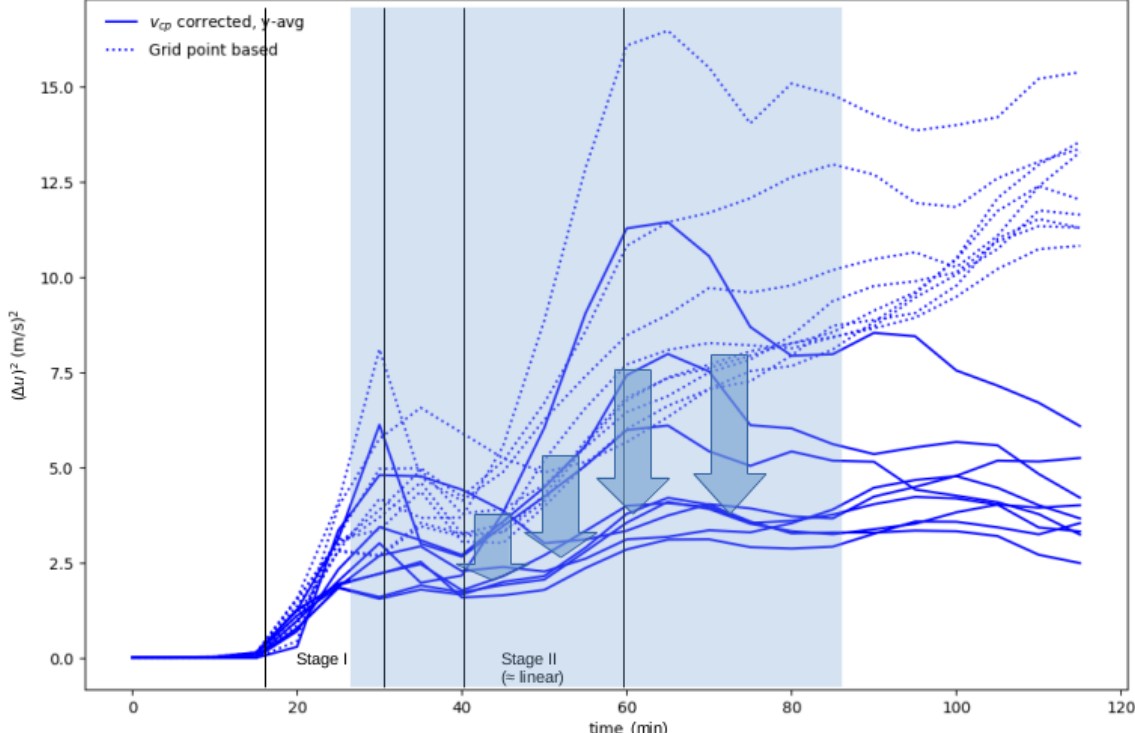

**Figure 11.** Error growth as measured by the ensemble variance in $u$, where each other member paired with the reference simulation. Both cold pool edge location corrected curves (solid lines) and grid point based comparisons (dotted) are displayed. Growth stages discussed in the text are also annotated in the Figure (see text) and the key interval discussed in the analysis is highlighted by blue shading.

stable. Moreover the ensemble members with large $(\Delta u)^2$ do seem to have a stable $v_{cp}$-corrected $(\Delta u)^2$ in the second hour. To conclude this section, two main stages of error growth have been identified, with no growth in between those two stages of

accelerated error growth and the second of the two reveals interesting near-linear relations with the cold pool edge propagation. More effective algorithms for cold pool edge detection could likely give even more insight into the quantitative role of the cold pool edge versus other processes in the squall line error growth.





## 4 Synthesis

### 4.1 Evolution of ensemble spread

#### 4.1.1 First amplification of ensemble spread: secondary convective initiation

The ensemble sensitivity analysis has demonstrated statistical correspondence between vertical velocity at location X in Figure 4 and flow patterns in the x-z plane elsewhere in time and space. In the section on the ensemble sensitivity analysis (Section 3.3.2) it has been shown that the updraft position after about 20 minutes of simulation time is already associated with differential vertical velocity after 30 minutes at this location X. The time evolution and spatial patterns of a through (resp. crest) in both the tracer distribution and the ensemble sensitivity confirm that the differential propagation of a gravity wave in terms of phase and amplitude is initiating a vertical velocity difference. This vertical velocity contrast is able to trigger extension of the convective cells a few km to the east after about 30-35 minutes - the secondary convective initiation in part of the ensemble and most notably in ENS-03. The secondary initiation directly impacts convective overturning: subsequently, ENS-03 and the reference simulation reveal a large difference in upward tracer flux of PT2 to levels above 4 and 6 km associated with the deep convection between 30 and 45, minutes by a factor of 2-3. This increased convective overturn also translates into increased average precipitation intensities (24%) and upper tropospheric divergence (38%), when averaged over the first 75 minutes of simulation time.

#### 4.1.2 Second phase in the evolution: cold pool acceleration

After the secondary phase of initiation happens (30-35 minutes), the upward mass flux within the convection is not only increased. The cold pool is also accelerated (around 35-45 minutes) to a speed that is later on maintained, as was suggested by the auto-correlation function of the cold pool edge location (Section 3.3.1). After about 50-55 minutes the cold pool edge location is highly self-correlated with its location at later times (beyond 100 minutes). The cold pool velocity is higher for ENS-03 with more convective overturn. One could anticipate that more convective overturn (updrafts) also leads to increased downdraft intensities. As the strengthening of updrafts is caused by eastward extension of the updrafts, one could also expect a consequent increase in downdraft area. Statistics reveal significant intra-ensemble correlation between the maximum downdraft flux as well as downdraft area at the level of maximum downdraft flux (30-75 min) with cold pool velocity (30-75 min). In other words: there is a link between the cold pool acceleration, overturn of the updrafts and downdraft effects.

Further support of the importance of cold pool acceleration in this stage is provided by the error growth curves in Section 3.4. This error growth curve starts to grow strongly in the gravity wave decorrelation and first convective initiation phase (20 minutes) and continues to do so during the first part of the secondary initiation phase (30-35 minutes). Directly after, the error growth stagnates for a short period (around 35-40 minutes). Curves where a correction is made for cold pool velocity clearly demonstrate less error growth in the subsequent phase than the uncorrected curves. By removing the cold pool propagation error from the squall circulation error curves, the error is reduced seriously, implying that much of the error is contained in the spread in cold pool position. Furthermore, among more than half of the ensemble pairs the $(\Delta u)^2$ in Figure 11 is reduced from





about $7m^2/s^2$ to $4m^2/s^2$ and in the other pairs significant reduction of the maximum is also seen. It implies some presence of feedback and interactions other than direct effects of cold pool propagation spread as source of ensemble variation, but an important mode of variability contained within the ensemble is certainly associated with the contrasts in cold pool velocity. The linear contribution of cold pool propagation is about as important to the errors as other non-linear feedback mechanisms in the second stage of error growth (40-60 minutes).

Summarizing, it could be argued that the secondary phase of initiation (30-35 minutes), which strongly affects consequent updraft fluxes, and subsequent cold pool acceleration (35-50 minutes) is the most important event for the ensemble spread. Once established, the cold pool velocity is maintained and is an important contributor to the squall line spread.

### 4.1.3   Common mode of variability and possible common driver

Our idea is that the squall line variability assessed in this study is strongly linked. There is probably common driver for much
of the squall line variability, including the cold pool acceleration to a stable value of $v_{cp}$. That means, the upward velocity after 30 minutes just ahead of the cold pool edge triggers the convective initiation, but might also affect the cold pool propagation: below a gravity wave crest (ENS-03), convergence was necessary as a compensation for upward motion. In connection with enhanced secondary convective initiation (due to stronger upward motion at X) this convergence pattern below the gravity wave crest implies that the cold air on the rear flank was likely accelerated a bit more eastward relative to the reference
member, in which no secondary initiation occurs. Simultaneously, the slight eastward acceleration should open up space for increased downdraft intensities and areas once the downdrafts are present. A pattern with extra space immediately available for downdrafts is also consistent with Hovmöller diagrams of the cold pool in ENS-03 and the reference simulation, because immediately after the secondary initiation the cold pool in ENS-03 occupied a larger area (supplement Figure S5). In addition, this is supported by the tracer evolution after 30-35 minutes. As the upward convective mass flux increases and there is slightly
more space for the downdraft to develop, downdraft area and mass flux are even more likely to increase. The statistical links between $w_{t=30}$ at X, $v_{cp}$ and downdraft area and mass flux are confirmed by Table 1. Moreover, the link is also suggested by the ensemble sensitivity analysis: slightly above the cold pool, a region of increased convergence (2-4 km above ground level, Figure 8) at the cold air boundary was found in association with high values of $w_{t=30}$ at location X. This region of increased convergence thereby increases the inflow of downdrafts below melting level at 2-4 km (Figure 10), exactly the height at which
the downdraft fluxes rapidly increase with decreasing height. In other words, the low level circulation around the cold pool is very likely increased by the secondary convective initiation.

The increased vertical overturn in both upward and downward mass fluxes is likely imposed on the squall line ensemble by a common main mode of variability. This mode of variability with increased convective overturn or equivalently intenser cells probably also induces much of the precipitation difference between ENS-03 and the reference simulation. Moreover, the mode
of variability probably induces some of the contrast in strength of convective overshooting into the stratosphere after about 60 minutes and the strength of mean upper tropospheric divergence (see Table 1). Therefore, a common driver of squall line variability has to exist, of which we have found $w_{t=30}$ at location X in Figure 4 as an early manifestation.

This $w_{t=30}$ at X is the earliest identified manifestation of the common mode of variability. It may be present before this





manifestation, hidden in the gravity wave variability within the ensemble, but that is beyond the scope of this study. The mode
of variability is responsible for much of the variability in $w_{t=30}$ at X, but lives on for the next 45-60 minutes. It covaries with
the variables shown in Table 1 and explains part of the variability in them. Furthermore it affects the circulation within the
squall line as revealed by the ensemble sensitivity analysis.

### 4.1.4 Approach of intrinsic limit?

Initial conditions vary very slightly - by under 2% for the shear layer depth and otherwise not at all between ENS-03 and the
reference simulation. The tiny initial difference implies that the mode of variability is very sensitive to initial conditions. Given
that the total variability in initial shear layer depth was 5%, it can be argued in line with Melhauser and Zhang (2012) (in
particular based on the illustration in their Figure 18) that there can occasionally be an intrinsic limit of predictability on time
scales of about an hour. A pattern very closely matching such behavior is exposed by our simulations and analysis. Identifying
this limit implies doing a perfect error assumption (see Selz, 2019).
Non-linear behavior in the evolution from initial conditions to $t = 30$ minutes would then explain why there is likely an in-
trinsic limit of predictability. The random realisation of gravity wave amplitude and phase after 30 minutes or just before that
time perhaps leads to random excitation of the identified mode of variability: Figure 4 suggests a possibility that amplitude and
phase of gravity waves is possibly directly responsible for $w$ after 30 minutes at location X. However, the latter is somewhat
speculative.
This non-linearity in the evolution between $t = 0$ and $t = 30$ minutes directly implies that characteristics of the initial condi-
tions cannot be monotonically linked with the conditions afterwards, including those during the secondary phase of initiation.
These resulting conditions also include the squall line circulation anomalies that were found in Section 3.3.2. Initial condition
perturbations are non-linearly linked with much of the squall line variability that occurs after 30-85 minutes. Therefore we
could apply the illustration of Figure 18 in Melhauser and Zhang (2012) to the findings and argue that intrinsic sensitivity is a
key implication.

### 4.1.5 Squall line circulation

The squall line circulation was very likely affected $w_{t=30}$ at X and so by the common mode of variability that we suggest.
It builds up after about 40-45 minutes of simulation time and relation with $w_{t=30}$ at X peaked after 55-60 minutes. The
anomalous squall line circulation pattern, as shown in Figure 8, resembles to some extent that of an enhanced circulation of
the jump updraft, downdraft and overturning updraft as in Moncrieff (1992). The figure suggests that the mode of variability
identified in this study is partly explained by intensification of this 2D circulation after 45-85 minutes of simulation time. If
this is true, corresponding downdraft and overturning updrafts are likely comparatively shallow.
During the second hour of simulations the traces of the identified mode of variability seem to disappear gradually. The gradual
dissipation of the mode of variability was identified by the ensemble sensitivity analysis between about 70 and 85 minutes, with
a new structure of $u$ variability appearing, which was opposing the earlier circulation signal in terms of zonal flow partly. In
combination with the flattening error curve in Figure 11 during the second hour this might suggest a sign of saturation, where





($\Delta u$) reaches about 4 m/s. That value is a significant portion of the actual $u$, but not yet near saturation (roughly halfway). There is a possibility that the identified common mode of variability covaries with a subhourly time scale imposed on the squall line by cyclic behavior during its growth as in McAnelly et al. (1997); Adams-Selin (2020a). That is very speculative and not

supported by traces of any other variables that were analysed, as well as beyond the scope of this study.

Next to the common mode of variability, other feedbacks can of course affect the squall line (circulation) and signals in retrieved diagnostics.

## 4.2 Discussion

While the squall line ensemble in this study is initiated with a cold pool damming structure and the convection starts to get

mature in the second half hour of simulation time, simulations last two hours in total. Furthermore simulations are restricted to a small domain around the squall line. This excludes part of the mature and the full dissipation stage from the analysis. However, the growth of tiny errors (0-20 min) to systematic variability (45-80 min) in the squall line circulation takes place on the time scales about an hour, so that the relevant window for squall line error growth is fully covered by the simulation time. The setting in our simulations is very similar to that in Adams-Selin (2020a, b). The main difference in environmental setting is

that this study has a reduced depth of the shear layer, to about half of their environment. Furthermore, the domain and geometry was different: squall lines in this study behave as "infinite" length squall lines. There is also similarity in the evolution in the sense that a discrete propagation event on small length scales is diagnosed, or at least a new cell growing ahead of our squall line. In our simulation this occurs a lot earlier: already after 30 minutes and just after the first phase of convective initiation. Furthermore the event occurs only a couple of km ahead of the squall line, as opposed to the clearer separation in Adams-Selin

(2020a, b); Fovell et al. (2006). One could wonder if our study is an actual discrete propagation event or just growth ahead of the squall line, but Figure 4 suggests it starts as a discrete propagation event. Quite soon our event is absorbed by the squall line itself.

Certainly, the discrete propagation event detected in Fovell et al. (2006) could not happen as closely ahead of the squall line as in our simulation, as the resolution was lower then and small updrafts are often at the minimum size possible in the simulation.

Our results definitely demonstrate that distinct convective initiation only about 3 km ahead of a squall line can easily be resolved at 200m resolution. With a resolution below 100m one could assume that events like discrete propagation could be simulated in LES when they happen at length scales of 500-1000m. This would require a higher output frequency as in Adams-Selin (2020a, b).

A degree of similarity is also shared with Weyn and Durran (2017), even though they use warm bubbles and slightly different

temperature, moisture and wind profiles with deeper shear of varying magnitude. In spite of their analysis mostly carried out in spectral space, their error growth curves of mesoscale convective systems can be compared to ours. The main difference is first of all that the simulations in this study take place in a smaller domain and at higher resolution for a shorter integration time. They run their simulations out to six hours, which leads to error saturation ($> 75\%$) after 4.2-6.0 hours. In the simulations presented here, error saturation is not reached. Moreover, the occurrence of some humps with stages of no error growth in the

error curves is shared with their Figure 8, but theirs appear relatively small in magnitude. However, for an assessment of error





saturation their study provides a much more suitable approach - the simulations here have not been integrated far enough and the metrics are not so suitable for that assessment.

Even though Weyn and Durran (2017) approach squall line predictability from a different perspective, this study provides additional groundwork on how squall line predictability may be extended very slightly only (on the order of an hour) when
initial conditions are very accurately known: initial conditions are non-linearly linked to the secondary phase of convective initiation and the secondary phase of initiation is highly sensitive to these initial conditions. On the other hand, the findings also suggest that some of the information on convective initiation can be stored as linear signals in the squall line relative circulation that live for time scales on the order of one hour, while feedbacks are secondary. That finding once more confirms the known ideas of shorter and shorter saturation time scales for smaller and smaller scales (Lorenz, 1969; Durran and Gingrich,
685    2014).

The ensemble in this study starts with highly but not fully 2D initial conditions, as part of a wide spectrum from highly simplified squall line studies (see Houze, 2004, for references) to more recent simulations of real cases coupled to the large scales (e.g. Melhauser and Zhang, 2012). Apart from the depth of the shear layer our initial conditions are practically the same as those in Adams-Selin (2020a). The high degree of 2D makes the variety of diagnostics applied here much more affordable
to assess, which is particularly beneficial in error growth studies. Even if not completely representative for many squall lines in the real atmosphere, convergent rolls often organise convection and occur along real squall lines. Insights in error growth provided here and in processes responsible for error growth are likely largely applicable along sections of real squall lines, especially those closely relating to findings of Melhauser and Zhang (2012). The idealisation only emphasizes signals, which is of benefit to the analyses.

The robustness of statistical results may appear somewhat questionable at the very first glance even after statistical tests, given that only two simulations are compared in Section 3.2 and given the ensemble size of just $n = 10$. Nonetheless, the very high correlations in Section 3.3.2 easily survive any significance tests with a null hypothesis that correlations found are rejected. These tests were developed with the aim to verify the robustness of the signal. Significance measures with which the ensemble sensitivity analysis survives the tests implied very high confidence in the statistical robustness of the signal. In other
words, larger ensembles would reveal the same signal as the 10 member ensemble, given the outcome of the statistical test. As confirmed by the uncorrected error growth curves, the signal of the main mode of variability is demonstrated to be well established and well separable from noise and non-linear contributions. These non-linear components consist of the feedbacks that do occur to a limited extent. The ensemble sensitivity analysis reveals the main mode of squall line variability, which is strongly related to the secondary phase of convective initiation with high confidence.

The relations found in this study indicate that cold pool acceleration after about 30 minutes of simulation time is very likely explained by the main mode of variability and hence by variability in gravity wave propagation. One could argue that the depth of heating profile immediately before cold pool acceleration can affect the vertical wave length and propagation speed of gravity waves of a certain wavenumber, in agreement with findings by Pandya and Durran (1996). A heating profile depth differing by 1 km would provide a variablity in cold pool propagation of about 8-10% for a 12 km deep troposphere and
explain an 8-10% difference in cold pool propagation speed. If a wave has a vertical wavenumber of $n = \frac{5}{2}$ over twice the





depth of the troposphere or a vertical wavelength of about 10 km, it will propagate at about 17 m/s. That wavelength is equal to wavenumber one in the layer between the surface cold pool and the tropopause. Against easterly near-surface winds that are on the order of 12-14 m/s these gravity waves would propagate eastward relative to this near-surface flow at low speeds of 3-4 m/s: the density current could surf on this wave. The 1.7 m/s difference in propagation velocity between ENS-03 and reference is consistent with this. Inspection of the precursor condensation rates after 25 minutes (before the gravity wave leads to secondary initiation) suggests a deviation in heating profile depth of at least 500 m between ENS-03 and the reference simulation, probably close to 1km. Furthermore, cloud top detections between the two deviate by 800m after 60-90 minutes of simulation time. In other words, there is support for this argument. The arguments are comparable to Grant et al. (2018) and references herein and Stechmann and Majda (2009). Whether the gravity wave propagation actually leads to differential cold pool acceleration is beyond the scope of our study. It would require a detailed minute by minute study for the 20-30 minutes interval for which output is not available. Computing and inspecting additional quantities such as the Scorer parameter would also be needed.

## 5   Conclusion

Comparison of a pair of idealised squall line simulations with nearly identical initial conditions and analysis of an idealised squall line ensemble demonstrates that some degree of linearity in error growth of the circulation is maintained during the second half hour of simulation, which is where the squall line grows mature. More specifically, a secondary phase of convective initiation that happens after 30-35 minutes of simulation time has been found as crucial for the evolution of ensemble spread. The mode of variability associated with this secondary phase of initiation is first revealed by the upward motion a few km ahead of the squall line after 30 minutes of simulation time, which is varies within the ensemble due to shifts in phase and amplitude of gravity waves triggered by the developing squall line itself. Leading to contrasts in secondary initiation of convection, the effect of this mode of variability on the squall line through upward tracer and mass transport, downward mass transport, cloud top height and variability in squall line circulation can be followed. Its effect is most clearly present after 55-60 minutes of simulation time. After almost an hour of existence this mode of variability gradually disappears (70-85 minutes).

In good agreement with Melhauser and Zhang (2012); Hanley et al. (2013) convective initiation does sensitively depend on environmental conditions in the presented ensemble of idealized simulations. The chain of interactions has been documented in much detail here. Following the crucial secondary initiation phase, a cold pool acceleration is seemingly coupled to the intensity at which mass is convected by updrafts and downdrafts, as revealed by the tracer transport. On the other hand, once established (35-45 minutes), the cold pool velocity $v_{cp}$ is maintained well throughout the next 30-45 minutes, as shown by Section 3.3.1. Variation in $v_{cp}$ explains much of the subsequent variance in cold pool perpendicular flow ($u$) and therefore a portion of it is explained by the mode of variability identified in this study. However, even when this mode of variation is removed by a correction (Figure 11), a substantial amount of ensemble variation is maintained. The latter implies that non-linear feedback mechanisms play at least some role in the idealised squall line ensemble after 45-60 minutes, but not the dominant role leading to non-linear growth.



Connected to the above, two stages with error growth are identified: those likely initiating due to decorrelating gravity wave
patterns (notably after 20-30 minutes) and the one constructing and expanding the main mode of squall line variability (40-50
minutes), which to a substantial extent depends linearly on $v_{cp}$ and the secondary phase of convective initiation (30-35 min-
utes).

Noteworthy, initial condition uncertainties lose their structure and less than 2% variation in initial top height of the shear layer
can result in convective intensities and overturned mass at opposite ends of an ensemble with 5% initial variation in top height
of the shear layer. An explanation for the loss of structure is likely that non-linear developments in the triggered gravity wave
signals (phase, amplitude, stage after 20-30 minutes) are a precursor to the phase of secondary initiation. Loss of uncertainty
structure within a narrow ensemble with squall line simulations is in good agreement with the picture that Melhauser and Zhang
(2012) draw for intrinsic and practical predictability at the end of their study. As in their study, it is found that a very small
subspace in initial conditions can provide the same ensemble spread as a wider subspace (just like among their simulations
classified as "good" and "poor"). Among one ensemble pair, initial condition variability even reduces to 0.006% in our study
and the whole ensemble spread applied is smaller than any realistic operational model can currently even describe. This means
the results presented here support the idea of an intrinsic limit of predictability.

*Code availability.* A readme file for the amended namelist file and main output data of simulations are available at https://tinyurl.com/
groot-tost-22 (Groot, 2022)

*Author contributions.* EG designed, conducted and composed the content of this study as part of his PhD project, with contributions from
and under the supervision of HT.

*Competing interests.* HT is also a co-editor of this journal. However, this does not represent a competing interest for this publication; there
are no further competing interests.

*Acknowledgements.* The research leading to these results has been done within the subproject 'A1 - Multiscale analysis of the evolution of
forecast uncertainty' of the Transregional Collaborative Research Center SFB / TRR 165 'Waves to Weather' funded by the German Research
Foundation (DFG). HT acknowledges additional funding from the Carl-Zeiss foundation.
The authors would like to thank Michael Riemer, Mirjam Hirt, Annette Miltenberger, Peter Spichtinger, Christopher Polster and Sören
Schmidt for their contributions through constructive feedback and/or suggestions to improve (sections of) the manuscript, methods and/or
improving the presentation in some figures.



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
