# Peer review of "Evolution of squall line variability and error growth in an ensemble of LES"

_EGUsphere, 2022_

## Author Response (AR3)

**Replies to all Referee Comments on "Evolution of squall line variability and error growth in an ensemble of LES " [1]**

Edward Groot, Holger Tost

December 23, 2022

**1 General**

As authors we would like to thank both referees very much for their thoughtful reading of (and extensive comments on) the manuscript.

**2 Referee 1: reply to RC1 [2]**

**2.1 High level comments**

We thank referee 1 for filtering out and formulating his pointwise summary, with which we also think most of the important outcomes have been covered [2].

*RC1: "(...) but rather a sharp eye and crisp tongue about what is important. The slow resurfacing at the end into Synthesis and then Conclusion sections (also rather long) does help to pull these key points and highlights out somewhat, but those key points could be even more polished into the Abstract for instance. (...) But those wrapup sections (4 and 5) come after a long slog of sometimes unclear prose in the late-middle (the very long sections 3.3 and 3.4)." and RC1:"Might the paper or at least sections 3.3 and 3.4 be cut by aspirationally 50% with no loss (and a gain of clarity) on the reader's part? Long stretches of text appear to describe figures not shown, without stating (not shown). "*

In general we do agree that the manuscript is obviously very lengthy at the moment. Therefore we will move considerable parts of Section 3.3 that are less crucial for the story line of the manuscript to the supplementary material, so that highly interested readers can access it easily, but that it does not add extra load to the main text for any other readers. Some parts will even be removed completely in the revised version.

*RC1: "Lines 19-20: The meaning of this result is that the perturbations chosen are in a non-essential field, but that even those differences grow (or explode). What does the word "intrinsic limit", lifted from some over-realm of philosophy it seems, really add to this idea?" and "480-onward: "error" —> "difference" "*

Regarding Section 3.4 [2]: referee 1 sees the importance of investigation of ensemble spread in LES simulations and our focus on convective dynamics, which is indeed the main point. However, these comments do suggest that its relation to "errors" in the representation of the atmosphere within a model, as well as theoretical limits of predictability, are seen as comparatively unimportant. Maybe we overinterpret referee 1's criticism by reading it in RC1 this way. From our point of view there is no need to judge on which perspectives are important to investigate or not - however, as authors we do think it is necessary to look at atmospheric modelling and predictability from this perspective, as [3, 4], [5, 6, 7] and other work by [8, 9, 10] has shown; this implies that Section 3.4 will not be as strongly modified as Section 3.3: even if "error growth" and "intrinsic limit of predictability" might never be important in studies of LES ensembles of convective systems, we choose to represent the error growth point of view in this work in line with works by colleagues in the field [3, 4, 5, 6, 7, 8, 9, 10] .

Moreover, terms as "intrinsic (limit of) predictability" and "errors" have been used very regularly in a set of recent predictability studies in the footsteps of Lorenz' work [e.g. [11]] and applied to various synoptic and convective scale case studies and more climatological studies [3, 8, 9, 6, 10, 7] (all of them subscales of the atmospheric system that Lorenz' work described). This was done in order to distinguish between practical

predictability in state-of-the-art NWP and explosive/exponential error growth with its implications for predictability from even smaller scales down to the smallest possible scales and/or larger scale differences in state of (much) smaller amplitude: the practical predictability representing current state-of-the-art (operational) global NWP and intrinsic predictability stemming from the smallest (scale/amplitude) errors. This means that it is not [2] ""intrinsic limit", lifted from some over-realm of philosophy" as formulated by Referee 1 in RC1, but rather directly addressed in (recent) meteorological studies. The cited papers (to the authors' opinions) depict why and how of given terms in the field of atmospheric modelling and predictability (of which not all atmospheric modellers will understandably be aware).

RC1: "From line 500 or so: the paper is getting rather long and verbose... can it be streamlined? If all the key results have been shown, why not gather them crisply and close?" and RC1: "but those key points could be even more polished into the Abstract for instance." and RC1: "how many words could be trimmed or eliminated without loss of meaning?"

On top of that the authors will pay attention to the writing style throughout the manuscript - reconsiderations in the discussion and conclusion sections will be done (including urgency of some paragraphs and sentences), which will be done in convolution with comments from RC2 [12].

RC1: "An unclear overanalysis in confusing statistical terms (terms like "source" and "error", and "autocorrelation" for intra-ensemble rather than temporal-lag correlations) were unhelpful or confusing, and it all gave few clear insights that aren't in the bullets above and in section 3.2."

Lastly, the use of the other "confusing terms" [2] is reconsidered (and mostly adjusted) for increased manuscript clarity. We hope to meet the expectations of referee 1 with the way we compact the (main) text and at the same time we hope to warranty completeness of the manuscript to any reader.

**2.2   Local comments**

We thank the reviewer for the local comments and will address them one by one in detail when updating the manuscript. Some specific comments (questions) will be adressed below.

- RC1: "L158: "interface height" — this is just a reference value in an analytic formula, which translates into shear strength on the 100m grid, right? This description was quite confusing. "
  We refer to Section 3.

- RC1: "L253: does your contouring routine treat \sloped features different from / sloped features?"
  There is no difference.

- RC1: "L399: "trough" and "crest" — what do these mean? Is this the u field, does it even show vertical displacements at all?"
  Yes, we see undulations, which coincide with perturbations in the $u$ and $w$ fields that show upward (downward) displacements and convergence/divergence patterns suggesting a through or crest.

- RC1: "L433: "compensating" — what does this word mean? It implies a big back-story in the authors' minds about how things are related and constrained, makes me nervous."
  There is not a really large "back-story", which means that we will choose other words.

- RC1: "L442-444: huh, how does removing all buoyant gridcells remove "gravity wave contributions from saturated parcels" ?"
  We think that the updated version will clarify this issue.

- RC1: "L408: "circulation" — what does it mean? We are looking at complicated multi-lobed structure of the u field."
  The convective overturn with main updraft and downdraft is meant with that. It includes a deeper overturn, but also shallower overturn and some entrainment (detrainment). In the revised manuscript circulation is sometimes replaced by (relative) $u$, flow anomalies/perturbation(s), etc. Essentially these refer to the structure found in Figure 8a/8b [1].

The comments that are not mentioned in the list have been addressed in 2.1 or are addressed below, in Section 3. Or in a few cases the figures or text in the manuscript will be updated or reconsidered for the revised version accordingly.

**3 Referee 2: reply to RC2 [12]**

**3.1 Major comments**

*RC2: "vertical grid spacing: while the authors invest a lot of computational resources into running an ensemble at a high horizontal resolution, the vertical grid spacing is relatively poor. The equidistant spacing of 100 m in the vertical is in my view inadequate to resolve the cold-pool dynamics and maybe also the melting layer properly. Especially as the cold pool plays a vital role for the further spread between ensemble members, a fine grid seems to be critical to resolve differences in the evolution of the cold pool, its interaction with the flow and the feedback onto the squall line dynamics. I suggest to rerun the control simulation with a vertically stretched grid and to document the differences in cold-pool dynamics with the equidistant grid."*

Regarding the vertical resolution, the selected choice of resolution has not directly been clarified and motivated within the manuscript.

We agree, that 100m vertical resolution may be inadequate to properly resolve the melting layer and also that cold pool dynamics might be affected by processes on smaller scales, such that they cannot be properly resolved with a layer thickness of 100m. However, the focus of this study was not on the cold pool dynamics, but the evolution of the ensemble spread of an idealised squall line with relatively small disturbances. Therefore, the aim is to identify aspects of convective systems and their dynamics that lead to variability and ensemble spread and thus how these aspects affect predictability, but not how smaller scale processes throughout the cascade are perfectly represented or not - an arbitrary length scale is setting the representation threshold for analysis, which in our case is somewhere near 1-2 km: cold pools can in principle just be represented.

Although we do see the point of referee 2 about vertical resolution, the concerns are from our point of view much lower. As the cold pool depth initiates at about 2.500 m and evolves to depths of 1-2 km (# grid cells 10-20), this appears to be close to but not yet at the critical low end of (cold pool) representation (about 7, but definitely at least 5 grid cells for any oscillations [13]) to the authors' opinion. Furthermore, the 100 m vertical resolution is very reasonable for representing squall lines and cold pool triggered systems in our opinion, based on studies by [14, 15, 16]. For the processes in the melting layer though, we would agree that the simulations are more or less *at* the critical low end. Indeed, comparison of the ensemble to a simulation at finer vertical resolution at low levels could therefore be seen as desirable, but as a short remark or paragraph in the supplementary material, which we will add.

That should be sufficient, as the representation of very subtle microphysical processes are not exactly within the scope of this study. Somewhat suboptimal, poorer representation of shallow layer microphysics and cold pool dynamics will possibly induce very similar biases in all ensemble members and we do not see any reason to assume that the ensemble spread is "blown-up" overly.

Having executed a simulation at 50m vertical resolution with improved cold pool representation, the reproduced copy of Figure 3 of [1] with the new simulation is very similar to Figure 3, with (in terms of details) intermediate behavior between the control simulation and ENS-03. Hence, it may be concluded that the cold pool propagation and dynamics has to fall somewhere in the middle of the ensemble envelope.

Of course the validity of the comment remains and we appreciate the referee for triggering our thoughts about this topic.

*RC2: "Please provide more information about the employed tracers, for example as a further subsection in section 2. How are they transported by the flow, in which way is the coupling to microphysics (sedimentation) and turbulence realized, what is the treatment of the tracers in the surface layer? The last point my be especially important as the lowest atmospheric layer is very deep, and without a careful treatment tracers may be stuck in the surface layer."*

Moving to the next major comment, the tracer is passive also with respect to clouds and precipitating particles. It is not sedimenting. The tendencies acting on the tracer are purely advective, plus the subgrid turbulence from the LES (will be mentioned in manuscript update). Some tracer mass can indeed get stuck in the boundary layer (probably meant with surface layer?), which was illustrated by the motion of 95-99% of the tracer mass when installing another tracer (not included in the manuscript). Once tracer mass escapes from the boundary layer or the layer below the inversion through updrafts, it can move away from that layer. This is what happened with the large majority of tracer mass initialised in front of the cold pool edge. These are the tracers included in the manuscript. Some of the mass of these two tracers could indeed return to the lower (near-surface/sub-inversion) layers after a fast, small scale circulation in the convective drafts. This has all been taken into account, but the tracers represented in the manuscript are those that undergo motion towards the convective region at $x = 0$ km (initially).

*RC2: "The Weisman and Klemp (1982) sounding is established and popular. Yet, it has been criticized for being very unstable and favourable for convection. For the case at hand this means that in all ensemble members a squall line develops. A profile that is less favourable for convective development the ensemble spread may be much larger, as some members may not be able to produce a vivid squall line."*

With regard to the comment about the Weisman and Klemp sounding: this is very true. The study we present assumes from a practical point of view that a line of strong updrafts initiates and this is certainly not always the case in an ensemble of simulations when squall lines might form, which has a large control on ensemble variability as well. Nevertheless, we decided to use this scenario, as the variability of interest is not whether a squall line develops, but how the spread in the potential squall line state evolves. Therefore, it represents a subsample space in potential squall line variability.

Of course, convective initiation by itself is a similarly interesting topic, but not in the scope of the presented study. From the other point of view (variability asssociated with the question "does deep convection initiate?"), earlier studies like [3] are offering a more insightful perspective and this study does not address that point of view. In other words, we argue that the preferred sounding depends on one's interests and thus point of view.

Since the case of no convective initiation would add a (0,0) point in many spaces of interest (i.e. no mass flux, precipitation and no flow anomalies as a result of deep convection), the extractable statistical signals are likely even highly influenced by cases of untriggered convection, burying the actual signal that is identified in this study. To the opinion the authors, looking at ensembles of convective systems in an integrated way that considers both perspectives however, could be a way to go for future theoretical studies, for which we have some ideas.

The initiation-conditional point of view will be pointed out more explicitly in the updated manuscript.

*RC2: "The reference simulation appears at one end of the spectrum, while I would have expected it somewhere in the middle of the ensemble. Do you have an explanation for this behaviour?"*

There is not supposed to be any preferred location with regards to the location of the reference simulation within the ensemble envelope. Initial condition perturbations are imposed on the thickness and top altitude of the shear layer, as described in the manuscript (and then translated to the model grid, as it will be addressed in more detail in the revised manuscript). Magnitudes of these perturbations vary from -5 to +1 or 2 % compared to the reference simulation, where the shear layer height in the reference run is 100%. The magnitude of the initial perturbations does not correlate with the perturbations at later times, as gravity waves are found to decorrelate those perturbation signals within the first 30 minutes.

When the convection is active and the cold pool accelerates, at about 30-40 minutes into the simulation, a perturbation structure that is maintained exists on a time scale of about an hour. After that, another episode of decorrelation starts.

The decorrelation phases destroy the structure in any initial condition perturbations and hence it is not known how any initial condition perturbation will evolve in this first stage of about 30 minutes (in our set-up). Predictions can only be built from the state after 30 minutes or so, towards states in the next hour. We tried to discuss this in the discussion section of the manuscript, but obviously, this section is going to be re-written in the updated manuscript to better elucidate this aspect.

**3.2    Minor/local commments**

For the updated manuscript, following is done in addition:

- *RC2: "subsection 2.1: please give more details about the formulation of microphysics, especially the treatment of the condensation process (via saturation adjustment ? ) and the evaporation process, as they will be crucial for the development of up- and downdrafts and thereby ensemble spread."*
  More microphysical details, including an extended description, references and code version, will be added in an updated 2.1

- *RC2: "Please provide more information on the perturbation of the initial conditions, especially on the perturbation of $z_i$. In which way is the perturbation of $z_i$ transferred to the atmospheric profile."*
  A short description of how perturbed interface heights are translated to winds at the model grid will be provided

- *RC2: "The top of the model domain is at 20 km, with a sponge extending down to 15 km. Taking a look at e.g. Figure 8 some of the convection seems to interact with the sponge already. Did you see*

*any signals of interaction of the convection with the sponge?"*
The convective cloud tops "live" at elevations of 10-13 km roughly. That means a very low fraction of convective clouds will exceed 13 km and the absolute limit that updrafts reach is at about 14 km. Similarly, the divergent upper tropospheric motion does stop at about 13 km too (Figure 8). There is a 1-2 km zone between the top of convective updrafts and the sponge layer. In the sponge layer itself only footprints of gravity wave motion are seen, as one would expect (sponge layer is there to damp these motions). Otherwise no clear dynamical effects are seen in this sponge layer. Therefore, there are no indications of (undesired) interactions of clouds and updrafts with the sponge layer.

- *RC2: "Figure 1: Please specify the computation of the parcel ascent. The red line seems to start at some elevated point. Please also remove the "(left)" statement."*
  Displayed is indeed the ascent of a parcel from the mixed layer at about 900 m altitude in the lower troposphere, as it will be described in the revised version.

- *RC2: "Section 3.1: please provide some detail about the computation of the radar reflectivity"*
  The computation is entirely based on CM1 output, which uses the specific humidity (cloud contents) in the grid boxes and the computation will be shortly described in the updated manuscript.

- *RC2: "section 3.3.1: there is some directional shear in the simulations given by the increasing v velocity component. The averaging over the y direction ignores this directional shear. In which way did you account for this?"*
  This minor asymmetry has not directly been accounted for, which is because the structure of the squall line remains highly linear/2D. In combination with the along-line spatial average, this is sufficient to extract the signals that are presented in [1]. Small corrections as suggested by [12] would have tiny impact on top of the spatial averaging.
  The meridional wind has been triggered to ascertain the manifestation of some 3D turbulence, as it has been described in the manuscript [1].

- *RC2: "Section 3.3.2: please give more detail on the ensemble sensitivity analysis. The section is impossible to understand without first taking a look into the cited papers. The 4th and 5th paragraph of the subsection is hard to follow, there is no figure supporting the statement "during the first 15 minutes of simulation time ..." and "After 15-20 minutes" "*
  Based on referee 1 [2], some of this unclear material will be moved to the supplementary material. The corresponding figures can also be found in the current version of the supplementary material.
  And: additional content on the ensemble sensitivity analysis will be provided in an updated version of the manuscript.

- *RC2: "Section 3.3.3 downdraft selection: by selecting only grid points that contain hydrometeors, the downdrafts where all rain has been evaporated will be disregarded. A better choice could be to increase the magnitude and/or to check for hydrometeors above."*
  The authors agree with the referee [12] that detecting columns in which evaporation takes place might be a better choice for detecting downdrafts. However, we think that this choice is not important to our results. We are not aware of any study comparing various algorithms with small differences like this for downdraft detection in a high resolution simulation of deep convection. A thorough comparison of detection algorithms would be a study on its own. In such a study a detailed comparison would definitely appropriate, including a test and evaluation of other aspects of the detection algorithm. The question of whether this suggestion would improve the downdraft detections is therefore well beyond the scope of this study and only considered relevant by the authors in a study where downdrafts play a (much) larger role.

- *RC2: "Line 461-467: I cannot follow the argumentation here. Judging from Figure 10 the downdraft mass flux at a height of 1.5-2km seems to show the largest variability. Please be more specific or rephrase."*
  The section describing the low level mass flux variability in downdrafts within the ensemble (Figure 10) will be updated, reconsidering the need for clarity.

- *RC2: "Line 590-593: I disagree with this statement. Downdrafts will carve their space, irrespective if there is space available or not, thereby killing updrafts."*
  These sentences will also be reformulated into something less suggestive, or we may consider removing a small portion completely (see also RC1 [2]).

Furthermore, the authors will consider and address the technical points of referee 2 to further improve the quality of manuscript.
We highly appreciate the contributions by referee 2 to improve the quality of the manuscript.

**4 Replies to Referee Comment 3 on "Evolution of squall line variability and error growth in an ensemble of LES " [1]**

As authors we would like to thank the reviewer for the thoughtful reading of (and extensive comments on) the manuscript. The comments are in our eyes is useful to further clarify the content of the study.

**4.1 High level comments [17]**

- *"The key messages of the study and the motivation for it aren't made sufficiently clear at the onset. Although there is a discussion of the mechanisms that will be studied, it is not clear at the onset what motivated the study (what questions were left open by previous studies). A clearer focus and more guidance would also help to streamline the results section."*

  We thank the reviewer for this comment. The authors hope that by addressing the aim of the paper and the gap in literature explicitly in a few sentences at the start of the introduction helps guiding the readers better.

- *"Several sections are difficult to read/interpret. Some sections could be reduced as they contain too much detail. A previous reviewer has already given some very helpful feedback here, which could be exploited further.*
  *- Section 2.4 is hard to interpret without more context.*
  *- Section 3 as a whole contains many numbers and details, which make it hard to read. The key messages are given in section 4. It would be better to only retain what is needed to support section 4. It is probably best to integrate these sections, so that results and their interpretation are presented together.*
  *- The supplementary material could likewise be reduced further (at this point, I have focused on the main text)."*

  In the revised manuscript, further context has been added to Section 2.4. It was not clear that statistical assessment was needed to test the obtained ensemble sensitivity for robustness, and now this has been clarified.
  The urgency of some results in Section 3 has been reconsidered and some have been removed from the main text. The reviewer wondered if Figure 5 was really needed, amongst others, and the authors believe that it is. It allows us to point out differences in cold pool propagation: Figure 5 in Section 3.2.2 provides a good bridge towards Sections 3.2.3 and 3.3.1. However, pointing out the upper tropospheric patterns here is probably indeed unnecessary (as widespread flow patterns are evaluated in 3.3.2 and all direct dynamical diagnostics of variability are wrapped up in Section 3.3.3). To canalise the results, discussion around Figure 5 specifically has been partly removed from the main material.
  Merging Sections 3 and 4 has been tried earlier (before the initial submission, in the early manuscript writing). However, for the same reason of canalising the focus of readers, we think that this solution would not be of benefit for the majority of readers. In a study with supporting results based on various analyses that lead to two main messages about characteristics of the error growth of the squall lines, a synthesis is in our eyes a better choice to bring the results together. The synthesis section has been appreciated in the earlier round of review RC1,RC2.

- *"The figures need some improvement (both in terms of presentation and clarity).*
  *It would be clearer to plot the tracer in each simulation (and only keep the difference plot if the differences are not clear from a direct comparison) in figures 4 and 5. In figure 5, I wonder if both tracers are needed to tell the story.*
  *For figure 6, why not simply show the trajectory over time for each of the simulations and use that as a starting point for discussion? The lag-correlation is harder to interpret.*
  *For figure 8 and 9, using equations rather than words in the text would make it clearer what precisely is plotted. In terms of presentation style, some axis labels are missing, and some text overlaps. Time units switch between minutes and hours."*
  Presentation of Figures has been improved in the updated manuscript. The Figure 4 now shows both tracer difference and actual tracer concentrations themselves for $t = 30$ minutes. This is indeed of benefit to the readers for a quicker interpretation of the Figure. However, once readers have seen the ordinary tracer concentration plot (provided now with the update), we believe that the readers can efficiently interpret the difference Figures in the following panels of Figure 4.
  At 30 and 35 minutes of simulation time, both the difference and the actual tracer concentration plot

give a satisfying picture according to the belief of the authors (which is because the tracer difference is caused mostly by displacements, where-ever a difference occurs). However, after 25 minutes, one is searching for patterns even more subtle (generally not shifts in location, but differences in local dilution of tracer concentrations). This means that the difference concentration is much more useful than the concentration of tracers, and the difference concentration is the best choice to visualise, in our eyes. Furthermore, guiding the reader by providing an X in the figure assists the reader on the way to recognise the most important patterns. If adding the X in the display of original concentrations, the authors believe that the Figure becomes less readable.

The necessity of Figure 5 has already been addressed in the reply to the previous main comment.

Regarding Figure 6, the authors believe that the added value of the lag correlation lies in its quantitative information on the cold pool edge location, which is essential as part of the overall quantitative analysis. However, we agree that an overview of both qualitative (actual cold pool edge locations) and quantitative (lag correlation of its location) representations of cold pool locations adds value to the overall story. Therefore, Figure 6 has been updated to include visualisations of both types of information. Furthermore, the previous version of Figure 6 (now: Figure 6, panel b) has been updated with an additional curve to supplement the information on the cold pool edge location.

The authors think that using equations to explain the computations done to obtain Figure 8 does not simplify interpretation of the Figure: it is more likely that this leads to a couple of additional sentences needed to explain symbols used in such an equation. These sentences would be easier to interpret than the currently included sentences - they would not differ substantially from the yet existing text. Furthermore, the explanation of the ensemble sensitivity analysis (the information at the start of Section 3.3.2) is supposed to explain the exact procedure of the calculation sufficiently well. The computational result of the procedure consists of local correlations and co-variance in $u_{avg}$ and $w_{loc}$, of which the former two are presented in Figure. We have added "co-" before variance, to emphasize that Figure 8 shows the co-variance within the ensemble of $u_{avg}$ associated with $w_{loc}$ at location of the cross in Figure 4b (updated manuscript) resulting from the ensemble sensitivity analysis. Computations needed for Figure 9 regard the definitions of updrafts and downdrafts earlier in the same Section. Based on the detailed comments of the reviewer (e.g. " *"(positive for updraft detection)"* → *remove*" [17]), the authors suppose that this definition is clear. Similarly, the authors assume that the procedure to extract analytical quantities displayed in Figure 9 is clarified in the accompanying text of Section 3.3.3. Lastly, flow has been averaged along the y-direction for Figure 9, but this averaging is not specific for Figure 9 and are supposed to be clear from other Sections of the manuscript (e.g. Sections 2.3, 3.3.2).

The problems with the lay-out of Figures 8 and 9 (e.g. "axis labels") have not been identified by the authors: axis labels ("x (km)"; "z (km)"; "u (m/s)") are present and neither have time indications in hours been found in Figures 8 and 9.

- " *Details about the differences between the simulations that form the ensemble are unclear.*
  *As I understand it, the simulations differ by the height of the zonal shear layer, but it is not clear which member corresponds to which interface height. Though the authors argue the interface height does not monotonically relate to e.g. "$w_{loc}$", it would still be good to order the simulations by it.*" The reviewer is right about the differences in initial conditions between the ensemble members. As the de-correlation from 0 to 30 minutes of simulation time is one of the two key points of the work, conveying this message has priority. Using coloring schemes in two Figures (6a and 10) where all ensemble members are addressed individually in the updated manuscript and by adding a colouring as an indication of the similarity between various members' initial conditions (thereby referring to $z_i$), further clarification is hopefully achieved. Furthermore, by adding the correlation value between initial conditions and $w_{loc}$ to the only table in the manuscript, the authors hope that the issue is completely clarified (after having been addressed by a previous reviewer ([2] and [12]), and subsequent first step in the clarification).

- "*In section 3.3.2, the precursors may not always be driving the target.*
  *For example, a higher precipitation flux may cause higher evaporation and then faster cold pool propagation. That said, faster propagation could indeed also lead to more intense convection. In this context, it may be worth looking at a paper by Alfaro (2017) in JAS "Low-Tropospheric Shear in the Structure of Squall Lines: Impacts on Latent Heating under Layer-Lifting Ascent".*"
  We thank the reviewer for pointing out this interesting paper ([18]). The occurrence of the de-correlation phase of initial perturbations within the first 30 minutes of simulations implies that the

initial shear layer top does not relate to eventual perturbations in latent heating and cold pool propagation directly, after the squall lines have formed. However, without the initial de-correlation phase as a result of gravity waves that is identified in our work, the work pointed out by the reviewer suggests that the low-level shear differences could directly affect the expected ensemble variability in terms of latent heating for example (by a small amount, almost 5%, interpreting [18]). This potential implication is now mentioned in the revised manuscript.

Nevertheless, as none of our diagnostics assesses the temporal evolution of the depth or strength of the shear layer and its variability later on in the simulations, we did not assess the specific relationship of the contemporary low level shear with cold pool propagation and latent heating rates.

**4.2  Detailed comments[17]**

- *"Check the text for compound (multi-word) adjectives, and hyphenate these: e.g. "three dimensional" → "three-dimensional"; "high resolution simulations" → "high-resolution simulations"."*
  Multi-word adjectives have been hyphenated in the updated manuscript.

- *"Remove/replace words that can be left out with no loss of information, e.g: "Presented diagnostics" → "diagnostics" ; "used scheme" → "microphysics scheme"; "The applied initial conditions" → "The initial conditions"."*
  The manuscript has been re-read in detail by the authors and has been checked for such issues. In general, in combination with the streamlining (as mentioned under the second bullet point of Section 4.1 of this reply), the authors hope that solving this issue has further clarified the manuscript and made it easier to digest for most readers.

- *"The subject "One" is overused in the text. I realise some authors try to avoid "we", but the use of the first person makes it clearer whether the authors agree with a line of thought or not."*
  This is correct for the previous version. During the re-read through the manuscript, we have addressed this issue and hopefully resolved it by replacing occurrences of "one" with first person and passive voice counterparts.

- *"Where two references are given outside parentheses, replace ";" by "and" (e.g. line 605)."*
  The authors believe that this issue has been resolved, by (where-ever appropriate) replacing those instances with a comma or "and".

- *"In several places in the introduction, the text is vague/general/unclear, for example:*
  *- Line 17: "Given the increasing computational resources"*
  *- Line 21: "It also includes the aspect of representation"*
  *- Line 25: "How squall lines depend on microphysics, shear and instability has been investigated rigorously by now, (e.g. Morrison et al., 2009; Grant et al., 2018; Adams-Selin, 2020a, b)." (also note the comma here)*
  *- Line 35: "This was the core feature of both sensitivity studies."*
  *- Line 69: "A sensitivity of these discrete convective cell was identified, which lead to a dependence of initiation on the active treatment of radiation."*
  As for the other issues, these occurrences of "vague text" (Reviewer 3, [17]) have been clarified according to the author's beliefs in the revised version: by a replacement with statements and descriptions that are more specific.

Line-by-line comments have all been addressed by replacing the words and sentences pointed out by the reviewer with corresponding suggestions provided by the reviewer, or by another similar revision of the textual details. In a few cases additional details have been provided in the manuscript, as requested by the reviewer. The authors think that the way most of these comments have been addressed and resolved is clear from the adjustments in the manuscript.

A few of the detailed comments need to be addressed specifically:

- *"Line 39: Mentioning the work of Lorenz (1969) here already would be beneficial."*
  The authors hope that this issue is addressed in accordance with the expectations of the reviewer. Although it was not 100% clear if the reviewer meant to include the point of shorter and shorter predictability time scales associated with smaller and smaller spatial scales identified in [11], we have assumed so and added this.

- *"Line 166: "Furthermore, the ensemble members all have slightly different boundary conditions, as controlled by their own evolution nearby/at the boundaries. The boundary conditions are solely based*

*on their conditions, with the first derivatives set to zero right at the boundary." → This is unclear, possibly what is meant is that the values at the boundary are different, even though the same type of boundary conditions is applied."*
This interpretation by the reviewer is correct and adjustments have been made accordingly.

- *" L 385: "would definitely pass the statistical significance test" → why not simply check it passes."*
  The authors have developed the statistical test to assess significance of the patterns within the squall line for the ensemble sensitivity analysis. Afterwards, the authors noticed that the same statistical test would have been passed ahead of the squall line by gravity wave signals, if the authors would have targeted at this area on beforehand. However, the authors believe that they should not test a hypothesis a posteriori, as statistical tests are generally specifically designed to have a definition on beforehand, suited to the appropriate requirements. Subsequently the test is executed.
  Unfortunately, the test was undertaken for only one of the two patterns identified with the ensemble sensitivity analysis and not for both. Therefore, we state the findings in the specific way we do, implying that if we had designed the test the same way for both gravity waves ahead of the squall lines and the flow within the squall lines on beforehand, both would pass the statistical test.

- *"- Line 24: "true convergence" → note that there may be convergence of bulk properties (see e.g. work by Wolfgang Langhans and others), even if there is no numerical convergence."* We agree that difference between local numerical convergence and statistical/bulk convergence is relevant for the paragraph about convergence and increasingly high-resolution simulations of squall lines: hence, we cite the paper [19].

**References**

[1] Edward Groot and Holger Tost. Evolution of squall line variability and error growth in an ensemble of les. *EGUsphere*, 2022:1–34, 2022.

[2] Referee comment 1 on "evolution of squall line variability and error growth in an ensemble of les", 2022.

[3] Christopher Melhauser and Fuqing Zhang. Practical and intrinsic predictability of severe and convective weather at the mesoscales. *Journal of the Atmospheric Sciences*, 69(11):3350–3371, 2012.

[4] Jonathan A. Weyn and Dale R. Durran. The dependence of the predictability of mesoscale convective systems on the horizontal scale and amplitude of initial errors in idealized simulations. *Journal of the Atmospheric Sciences*, 74(7):2191 – 2210, 2017.

[5] L. Bierdel, T. Selz, and G.C. Craig. Theoretical aspects of upscale error growth through the mesoscales: an analytical model. *Quarterly Journal of the Royal Meteorological Society*, 143(709):3048–3059, October 2017.

[6] Tobias Selz. Estimating the intrinsic limit of predictability using a stochastic convection scheme. *Journal of the Atmospheric Sciences*, 76(3):757–765, 2019.

[7] Tobias Selz, Michael Riemer, and George Craig. The transition from practical to intrinsic predictability of midlatitude weather. *Journal of the Atmospheric Sciences*, 2022.

[8] Fuqing Zhang. Dynamics and structure of mesoscale error covariance of a winter cyclone estimated through short-range ensemble forecasts. *Monthly Weather Review*, Oct 2005.

[9] Fuqing Zhang, Naifang Bei, Richard Rotunno, Chris Snyder, and Craig C. Epifanio. Mesoscale predictability of moist baroclinic waves: Convection-permitting experiments and multistage error growth dynamics. *Journal of the Atmospheric Sciences*, 64(10):3579–3594, October 2007.

[10] Fuqing Zhang, Y. Qiang Sun, Linus Magnusson, Roberto Buizza, Shian-Jiann Lin, Jan-Huey Chen, and Kerry Emanuel. What is the predictability limit of midlatitude weather? *Journal of the Atmospheric Sciences*, 76(4):1077 – 1091, 2019.

[11] Edward N. Lorenz. The predictability of a flow which possesses many scales of motion. *Tellus*, 21(3):289–307, 1969.

[12] Referee comment 2 on "evolution of squall line variability and error growth in an ensemble of les", 2022.

[13] William C. Skamarock. Evaluating mesoscale nwp models using kinetic energy spectra. *Monthly Weather Review*, 132(12):3019–3032, 2004.

[14] George H. Bryan, John C. Wyngaard, and Michael J. Fritsch. Resolution requirements for the simulation of deep moist convection. *Monthly Weather Review*, 131:2394–2416, 2003.

[15] George H. Bryan and Hugh Morrison. Sensitivity of a simulated squall line to horizontal resolution and parameterization of microphysics. *Monthly Weather Review*, 140(1):202 – 225, 2012.

[16] Leah D. Grant and Susan C. van den Heever. Cold pool dissipation. *Journal of Geophysical Research: Atmospheres*, 121(3):1138–1155, 2016.

[17] Anonymous referee 3 comment on "evolution of squall line variability and error growth in an ensemble of les", 2022.

[18] Diego A. Alfaro. Low-tropospheric shear in the structure of squall lines: Impacts on latent heating under layer-lifting ascent. *Journal of the Atmospheric Sciences*, 74(1):229 – 248, 2017.

[19] Wolfgang Langhans, Juerg Schmidli, and Christoph Schär. Bulk convergence of cloud-resolving simulations of moist convection over complex terrain. *Journal of the Atmospheric Sciences*, 69(7):2207 – 2228, 2012.